# QRIM: QUANTUM ROBUST INNER MINIMIZATION FOR REINFORCEMENT LEARNING

## ABSTRACT

Reinforcement learning (RL) often fails when faced with unexpected environmental changes that were unseen during training. Robust reinforcement learning (RRL) tackles this challenge by optimizing policies against the worst-case scenario defined within an uncertainty set. However, RRL remains impractical due to the cost of the Max-Min optimization, where it suffers from the query complexity for exhaustively finding the worst-case (dubbed 'Min') within the uncertainty set $\mathcal{U}$, i.e., $\mathcal{O}(|\mathcal{U}|)$. By viewing this via a lens of quantum perspective, we raise a pivotal question: *If we can query from the environment with quantum superpositions, is it possible to accelerate the Max-Min optimization of RRL?* Our answer is 'Yes'. Our method, called quantum robust inner minimization (QRIM), encodes the uncertainty set with quantum superposition and amplifies low-return cases, thus enabling RL for solving the robust (i.e., worst-case) Bellman equation. Importantly, QRIM achieves a quadratic speed-up in query complexity without altering the outer RL pipeline, i.e., $\mathcal{O}(\sqrt{|\mathcal{U}|})$. Validated through classical simulations to real quantum hardware execution, QRIM learns more robust policies for unseen task variations than classical RL methods, while achieving a quadratic reduction in query complexity compared to classical RRL methods.

## 1 INTRODUCTION

Reinforcement learning (RL) optimizes agents' actions in a given environment by maximizing the expected return, where the parameterized rule for taking an action is called *policy*. In practical scenarios, RL often suffers from a drastic performance degradation when deployed in a real-world environment, which shows a significant gap with simulations. Robust reinforcement learning (RRL) primarily aims to address this mismatch by introducing an *uncertainty set*, which covers possible environment variations, and maximizing the policy's performance (dubbed 'Max') under the worst-case scenario (dubbed 'Min'), so-called 'Max-Min' optimization. By assuming the state-action rectangular uncertainty, the robust value function satisfies robust Bellman equations, and it permits agents to find a principled planning procedure in a finite space (Iyengar, 2005; Nilim & Ghaoui, 2005; Wiesemann et al., 2013).

The existing RRL algorithms approximate their robust policy solutions by applying the robust Bellman operator at each sampled state–action pair, iteratively. In each step of the robust operators, RRL requires solving the inner minimization via recognizing the worst-case transition within the uncertainty set, i.e., $\mathcal{U}$, and estimating the corresponding Bellman backup. The crucial problem is that when the uncertainty set expands or a tighter confidence guarantee of the approximation is required, the inner minimization process, aimed at searching for the worst-case, rapidly inflates the computational and query (i.e., sample) complexities, whose complexity can be approximated as $\mathcal{O}(|\mathcal{U}|)$. This becomes the primary bottleneck of practical usage of RRL, regardless of whether the RL architecture follows value-based or policy gradient.

We aim to view this through a quantum perspective to anticipate a novel remedy for the bottleneck. Let us begin with a pivotal question: *If queried from the environment with quantum superpositions, can we accelerate the inner minimization process of RRL?* Our answer is 'Yes'. To represent a given environment with superposition, we borrow the concept of *quantum-accessible environment* (Wang et al., 2021), which is an interface that prepares a quantum superposition over possible next states with amplitudes equal to the square roots of their transition probabilities. Specifically, for a

state-action pair $(s, a)$ and each uncertain case in $\mathcal{U}$, a quantum-accessible environment provides *unitaries* $U_P$, $U_R$, and $U_V$ with quantum superpositions over next states and loads the associated rewards and values (see Definition 2; subscripts $P$, $R$, $V$ simply mean probability, reward, and value); thereby coherently encoding the one-step return on an ancilla for amplitude estimation. Built on it, our proposed quantum robust inner minimization (QRIM) estimates the amplitude of the worst case with the minimal reward within the uncertainty set by employing a quantum minimum finding algorithm, whose query complexity is $\mathcal{O}(\sqrt{|\mathcal{U}|})$, which is significantly lower than the conventional worst-case exhaustive search (i.e., $\mathcal{O}(|\mathcal{U}|)$ of RRL). We emphasize that a quantum approach can alleviate the impracticality of sample complexity for classical robust policy training, shedding light on a novel benefit of rethinking classical RL problems from a quantum perspective.

To intuitively illustrate the speed-up gains of our approach, we present a toy example inspired by a local snapshot of the FrozenLake environment in Fig. 1, which is a navigation problem. Consider an agent facing an uncertainty set of size $N = 4$ (e.g., four possible wind directions), where one direction leads to a hole representing the worst-case scenario with a negative reward $(-1)$, while others are safe $(0)$. In the classical robust RL framework (Fig. 1, at the top box), the agent must sequentially simulate each of the 4 transition possibilities to identify the minimum reward. This exhaustive linear scan results in a query complexity of $\mathcal{O}(N)$. In contrast, our proposed QRIM (Fig. 1, at the bottom box) encodes these scenarios into a quantum superposition, visualized as transparent grids processed simultaneously. QRIM identifies the worst-case outcome with a quadratic speed-up, requiring only $\mathcal{O}(\sqrt{N})$ queries, which is 2.

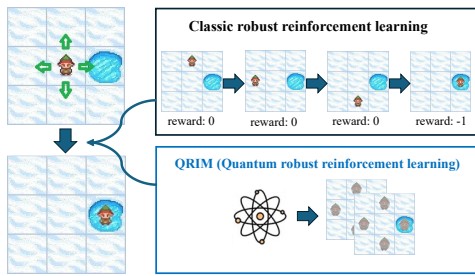

Figure 1: **Query complexity in a Toy MDP** ($N$=4). **(Top)** Classical RRL requires sequential scans ($O(N)$). **(Bottom)** QRIM uses superposition for parallel search, identifying the worst-case in $O(\sqrt{N})$ steps.

Our contributions are summarized as follows: First, we propose QRIM, a quantum approach that accelerates inner minimization of the RRL problem by encoding uncertainty sets in superposition and amplifying low-return cases. In addition, we provide theoretical guarantees on QRIM's accuracy and query complexity, showing a quadratic speed-up over classical RRL methods. Lastly, based on a classical emulator for quantum-access oracle, we confirm that QRIM learns robustness policies with significantly reduced total query counts, e.g., it achieves $\times 2.17$ out-of-distribution test reward over classical RL, and $-79.5\%$ query reduction compared to classical RRL in the CartPole benchmark.

## 2 RELATED WORK

### 2.1 ROBUST REINFORCEMENT LEARNING

Robust Markov decision processes (MDPs) introduce rectangular uncertainty sets at the state-action level and show that the associated robust Bellman operator is a contraction with a unique fixed point in finite spaces (Iyengar, 2005; Nilim & Ghaoui, 2005); these results underpin robust planning by value or policy iteration (Wiesemann et al., 2013). There exist two dominant families of RRL methods: value-based learners that approximate targets induced by the robust Bellman operator (rooted in (Iyengar, 2005; Nilim & Ghaoui, 2005)) and policy-gradient learners that derive gradients for the Max-Min objective under rectangular uncertainty (Wang & Zou, 2022; Kumar et al., 2023). In both cases, each update at a sampled $(s, a)$ requires the *inner minimization* (so-called worst-case Bellman backup), which becomes the dominant cost as the uncertainty resolution grows. To mitigate this cost, adversarial training learns a disturbed policy to intentionally expose agents to the failure cases (RARL) (Pinto et al., 2017), and ensemble methods optimize lower quantiles over environment samplings (EPOpt) (Rajeswaran et al., 2017); these approaches aim to mimic the undesired defectives in training yet avoid the explicit worst-case backup. A complementary direction replaces the uncertainty set with the set-valued kernels by distributional ambiguity balls bounded by total variation or Wasserstein distance, yielding geometry-specific convex inner problems and generalization bounds (Xu & Mannor, 2012; Yang, 2017; Chen et al., 2019) particularly with accelerated solvers

for the Wasserstein case (Grand-Clément & Kroer, 2021; Yu et al., 2023). Nonetheless, evaluation on fine-grained uncertainty grids still requires $O(|\mathcal{U}|)$ scans over candidate scenarios.

## 2.2 QUANTUM REINFORCEMENT LEARNING

Quantum reinforcement learning (QRL) has developed along two main streams (Meyer et al., 2022). The first employs variational quantum circuits (PQCs) to parameterize policies or value functions, trained with classical optimizers. Jerbi et al. (Jerbi et al., 2021) introduced PQC-based policies and demonstrated their trainability on RL environments with a few qubits, highlighting potential representational benefits. In parallel, Kim et al. (Kim et al., 2025) explored PQC policies in applied control settings, emphasizing how circuit design choices and task structure influence learning stability and performance. The second direction shifts its focus to an environment perspective, whose main goal is to construct an oracle-like access to a quantum-accessible environment (or quantum generative model). The core benefits are the accelerated planning or evaluation by composing coherent state preparation with standard quantum primitives. As a key enabler for estimating the amplitude of a quantum-accessible environment, Wang et al. (Wang et al., 2021) analyze algorithms that leverage quantum amplitude estimation (QAE) to reduce the dependence on the target accuracy compared to classical Monte Carlo estimation. QAE estimates a bounded mean within additive error $\varepsilon$ using $\mathcal{O}(\varepsilon^{-1})$ queries compared to the $\mathcal{O}(\varepsilon^{-2})$ calls needed by classical Monte Carlo (Brassard et al., 2002).

Despite these advances, robustness has not been a primary focus in QRL: variational approaches study the expressivity and learnability of quantum policies, while oracle-based works accelerate classical planning or exploration; neither directly targets the Max-Min structure of robust MDPs or the repeated inner minimization that overloads the training. In contrast, we explicitly implement robust reinforcement learning under rectangular uncertainty: our method, dubbed QRIM, computes robust Bellman backups across a discretized uncertainty set and identifies the worst case, thereby realizing robustness within a quantum-access framework.

## 3 PRELIMINARIES

As preliminaries, let us begin with providing standard notations for RL and proceed to formally describe robust MDPs with rectangular uncertainty sets. Afterward, for a quantum perspective, let us specify the formal definition of a quantum-accessible environment and its fundamental propositions.

### 3.1 RL BACKGROUND

Let $\mathcal{S}$ and $\mathcal{A}$ be the state and action spaces, and let $P$ denote the transition kernel (i.e., for each $(s, a)$, $P(\cdot \mid s, a) \in \Delta_{\mathcal{S}}$, where $\Delta_{\mathcal{S}}$ is the set of probability measures over $\mathcal{S}$). An MDP $(\mathcal{S}, \mathcal{A}, P, r, \gamma)$ with discount factor $\gamma \in (0, 1)$ evolves with a state transition from step $t$ to $t + 1$, i.e., $s_{t+1} \sim P(\cdot | s_t, a_t)$, and results a reward, i.e., $r_t = r(s_t, a_t, s_{t+1})$. A stationary policy $\pi(\cdot | s)$ induces value functions:

$$V^\pi(s) = \mathbb{E}_\pi\Big[\sum_{t \geq 0} \gamma^t r_t \,\Big|\, s_0 = s\Big], \qquad Q^\pi(s, a) = \mathbb{E}_\pi\Big[\sum_{t \geq 0} \gamma^t r_t \,\Big|\, s_0 = s, a_0 = a\Big].$$

The Bellman operators are formulated as follows:

$$(T^\pi V)(s) = \mathbb{E}_{a \sim \pi} \mathbb{E}_{s' \sim P}\big[r(s, a, s') + \gamma V(s')\big], \quad (T_* V)(s) = \max_{a \in \mathcal{A}} \mathbb{E}_{s' \sim P}\big[r(s, a, s') + \gamma V(s')\big].$$

Both are $\gamma$-contractions in $\|\cdot\|_\infty$, yielding unique fixed points and supporting value/policy iteration in finite spaces. With a function approximation, *value-based* learners estimate sampled Bellman targets, while *policy-gradient* learners optimize $J(\theta) = \mathbb{E}[V^{\pi_\theta}(s_0)]$ via computing gradients:

$$\nabla_\theta J(\theta) = \mathbb{E}_{(s,a) \sim d^{\pi_\theta}}\big[\nabla_\theta \log \pi_\theta(a|s) \, A^{\pi_\theta}(s, a)\big], \qquad A^\pi = Q^\pi - V^\pi.$$

Especially to mitigate the high variance of policy gradients, actor–critic methods introduce a learned value function (critic) to stabilize updates. In value-based and actor–critic methods alike, learning relies on estimating accurate one-step targets $r(s, a, s') + \gamma \widehat{V}(s')$ for critic updates at sampled states.

## 3.2 ROBUST MDPs UNDER RECTANGULAR UNCERTAINTY

In standard MDPs, the transition kernel $P(\cdot \mid s, a)$ and reward $r(s, a, s')$ are assumed to be fixed. Robust reinforcement learning instead considers a set of possible transition–reward pairs, called an *uncertainty set*, which reflects possible variations in transition kernels and rewards. At each state–action pair $(s, a)$, the adversary may choose one candidate model $(P_i, r_i)$ from this set, resulting in different one-step transition outcomes. To make this precise and to fix notation, we formalize the uncertainty at $(s, a)$ by an uncertainty set of transition–reward pairs and specify the notion of *rectangularity* (independent selection across state–action pairs) as follows.

**Definition 1** (Uncertainty set and rectangularity (Iyengar, 2005; Nilim & Ghaoui, 2005)). *Let $\mathcal{U}_{sa} \subseteq \{(P, r) : P(\cdot \mid s, a) \in \Delta_{\mathcal{S}}, \ r : \mathcal{S} \to \mathbb{R}\}$ be a nonempty set of one step laws, where $(P, r)$ specifies both the transition kernel $P(\cdot \mid s, a)$, the reward function $r(s, a, \cdot)$ and $\Delta_{\mathcal{S}}$ is the set of probability measures over $\mathcal{S}$. Rectangular uncertainty allows the adversary to choose an element of $\mathcal{U}_{sa}$ independently at each $(s, a)$.*

For bounded rewards, the robust optimality operator becomes a Max–Min problem:

$$(T_{\mathrm{rob}} V)(s) \ = \ \max_{a \in \mathcal{A}} \ \min_{(P, r) \in \mathcal{U}_{sa}} \ \mathbb{E}_{s' \sim P(\cdot \mid s, a)} \big[ r(s, a, s') + \gamma V(s') \big]. \tag{1}$$

**Proposition 1** (Contraction and fixed point (Iyengar, 2005; Nilim & Ghaoui, 2005)). *$T_{\mathrm{rob}}$ is a $\gamma$-contraction in $\| \cdot \|_{\infty}$ with unique fixed point $V^*$. Any policy greedy with respect to $Q^{\mathrm{rob}}(\cdot, \cdot; V^*)$ is robust-optimal, where*

$$Q^{\mathrm{rob}}(s, a; V) \ = \ \min_{(P, r) \in \mathcal{U}_{sa}} \ \mathbb{E}_{s' \sim P(\cdot \mid s, a)} \big[ r(s, a, s') + \gamma V(s') \big]. \tag{2}$$

For feasibility, we often discretize each $\mathcal{U}_{sa}$ into $N$ candidates $\{(P_i, r_i)\}_{i=1}^{N}$, such as grids over parameters, ensembles, or samples from ambiguity balls (Xu & Mannor, 2012). The inner minimization for each candidate is then written as

$$g_i(s, a; V) \ := \ \mathbb{E}_{s' \sim P_i(\cdot \mid s, a)} \big[ r_i(s, a, s') + \gamma V(s') \big], \qquad Q^{\mathrm{rob}}(s, a; V) = \min_{i \in [N]} g_i(s, a; V). \tag{3}$$

The practical bottleneck is identifying $Q^{\mathrm{rob}}$ and accurately evaluating $g_i$ from many $(s, a)$ queries.

## 3.3 QUANTUM-ACCESSIBLE ENVIRONMENT AND BASIC PROPOSITIONS

We build upon the quantum-accessible environment in RL planning (Wang et al., 2021) and the basic propositions for amplitude estimation and minimum searching in quantum settings. We use standard quantum computing Dirac notation: kets (e.g., $|i\rangle$) denote computational-basis qubit states, tuples such as $|x, y, z\rangle$ abbreviate tensor products. Additional explanation of the notation is in Appendix B.

**Definition 2** (Quantum-accessible environment). *For classical inputs $(s, a)$ and a basis index $i \in [N]$ of a discretized environment, a quantum-accessible environment provides unitaries as follows:*

$$\text{(i) Transition sampler } U_P : \ |s, a, 0\rangle \mapsto \sum_{s'} \sqrt{P_i(s' \mid s, a)} \, |s, a, s'\rangle,$$

$$\text{(ii) Reward oracle } U_R : \ |s, a, s', 0\rangle \mapsto |s, a, s', \ r_i(s, a, s')\rangle,$$

$$\text{(iii) Value oracle } U_V : \ |s', 0\rangle \mapsto |s', V(s')\rangle,$$

*together with standard reversible arithmetic (add/multiply/compare) on fixed-precision registers.*

**Proposition 2** (Quantum amplitude estimation (Brassard et al., 2002)). *Given a unitary $A$ that prepares $\sqrt{\mu} |1\rangle + \sqrt{1 - \mu} |0\rangle$ on an ancilla (possibly entangled), there exists an algorithm that, with $M$ coherent queries to $A$ and $A^{\dagger}$, outputs $\widehat{\mu}$ such that $|\widehat{\mu} - \mu| \leq c/M$ with probability at least $2/3$ for a universal constant $c$. Median-of-means boosting yields, for any $\varepsilon, \delta \in (0, 1)$, an estimate with $|\widehat{\mu} - \mu| \leq \varepsilon$ and failure probability $\leq \delta$ using $\mathcal{O}(\varepsilon^{-1} \log(1/\delta))$ queries.*

**Proposition 3** (Quantum minimum finding (Dürr & Høyer, 1996)). *Let $f : \{1, \ldots, N\} \to \mathbb{R}$ be accessible through an oracle that, given two indices, can mark (or compare) the smaller value with error probability at most $\eta < \frac{1}{2}$. Then there is a procedure that returns $\widehat{i} \in \arg \min_i f(i)$ with failure probability $\leq \delta$ using $\tilde{\mathcal{O}}(\sqrt{N} \log(1/\delta))$ mark/value queries; the overhead from $\eta$ can be absorbed into the polylog factor by repetition.*

In our application, we set $f(i) = g_i(s, a; V)$. Minimum searching can be implemented by running the quantum amplitude estimation described by Proposition 2. We provide explicit circuit constructions of these oracles for the RL environments used in our experiments in Appendix D, illustrating how the quantum-accessible environment can be realized in principle.

## 4 METHODOLOGY

Before describing our methodology dubbed QRIM, we bound rewards and reformulate the robustness objective $g_i(s, a, V)$ with a quantum amplitude estimation.

**Assumption 1** (Bounds for normalization). *Rewards satisfy $R_{\min} \leq r_i(s, a, s') \leq R_{\max}$, and $V$ is bounded. Set $V_{\max} = R_{\max}/(1 - \gamma)$ and $V_{\min} = R_{\min}/(1 - \gamma)$, hence*

$$L := R_{\min} + \gamma V_{\min}, \qquad U := R_{\max} + \gamma V_{\max},$$

*so $r_i(s, a, s') + \gamma V(s') \in [L, U]$ for all admissible outcomes.*

**Proposition 4** (Amplitude encoding of $g_i$). *Under Definition 2 and Assumption 1, one can implement a state-preparation unitary $\mathcal{G}_i(s, a)$ for fixed $(s, a)$, which prepares*

$$\mathcal{G}_i(s, a)|0\rangle = \sum_{s'} \sqrt{p_i(s'|s, a)} |s'\rangle |\mathrm{aux}\rangle \left(\sqrt{u} |1\rangle + \sqrt{1 - u} |0\rangle\right), \quad u = \frac{r_i(s, a, s') + \gamma V(s') - L}{U - L},$$

*where $u \in [0, 1]$. Let $\mu_i = \mathbb{E}[u]$, then $\mu_i = (g_i(s, a; V) - L)/(U - L)$ and $g_i = L + (U - L)\mu_i$.*

Based on them, we here describe the inner minimization routine called QRIM, whose interface is compatible with value-based and policy-gradient updates.

### 4.1 MAIN ALGORITHM

As described in Alg. 1, our quantum routine repeatedly returns estimator $\widehat{Q}^{\mathrm{rob}}(s, a; V)$ for state-action samples, thereby converging to the robust policy. The essential part is executed by QRIM, which computes $\widehat{Q}^{\mathrm{rob}}(s, a; V)$ for each state-action pair.

---

**Algorithm 1** Quantum-Accelerated Robust Learning (high level)

---

1: Initialize policy/value parameters; expose the classical value oracle $U_V$.
2: **repeat**
3:     Collect data (on-policy for policy-gradient, or from replay for value-based).
4:     **for** each sampled $(s, a)$ **do**
5:         $\widehat{Q}^{\mathrm{rob}}(s, a; V) \leftarrow \mathrm{QRIM}(s, a; V)$             ▷ Alg. 2
6:     **end for**
7:     Form targets/advantages using $\widehat{Q}^{\mathrm{rob}}$ and update with a standard optimizer.
8: **until** convergence or budget

---

### 4.2 INNER MINIMIZATION: QRIM

For each discretized candidate environment $(p_i, r_i)$ at a given $(s, a)$, define

$$g_i(s, a; V) = \mathbb{E}_{s' \sim p_i}[r_i(s, a, s') + \gamma V(s')], \qquad Q^{\mathrm{rob}}(s, a; V) = \min_{i \in [N]} g_i(s, a; V).$$

Under the quantum-access interface (**Definition** 2) and normalization (**Assumption** 1), **Proposition** 4 allows QRIM to provide a unitary $\mathcal{G}_i(s, a)$ whose ancilla amplitude has mean $\mu_i = \frac{g_i - L}{U - L}$. We utilize amplitude estimation (**Proposition** 2) as a building block and compose it with quantum minimum finding (**Proposition** 3).

QRIM exposes two tolerances of minimum finding and amplitude estimation:

- $\varepsilon_{\mathrm{cmp}} > 0$ (comparison tolerance): additive accuracy used when comparing two candidates inside quantum minimum finding. Each comparison invokes QAE on $A_i$ and $A_j$ to accuracy $\varepsilon_{\mathrm{cmp}}$, deciding whether $g_i \leq g_j$ with high confidence.

- $\varepsilon_{\mathrm{est}} > 0$ (estimation tolerance): additive accuracy used when re-estimating the winner, (i.e., worst-case) $\widehat{i}$ at the end, producing the robust value $\widehat{Q}^{\mathrm{rob}}(s, a; V)$.

More sampling of coherent queries is required for reducing the uncertainties (**Proposition** 2). Alg. 2 shows the algorithmic description of QRIM. Briefly, we run a QAE (line 2) and search for the worst-case (from line 5 to 9). For the estimated pair of cases with tolerance $\varepsilon_{\mathrm{cmp}}$ (line 6), we run Dürr–Høyer minimum finding among the samples (line 9). Finally, we return the worst-case (lines 10, 11).

---

**Algorithm 2** QRIM: Quantum Robust Inner Minimization at $(s, a)$

---

**Require:** Oracles $\{\mathcal{G}_i(s, a)\}_{i=1}^N$; tolerances $(\varepsilon_{\mathrm{cmp}}, \varepsilon_{\mathrm{est}})$; failure prob. $\delta$
1: **function** ESTIMATE$(i, \varepsilon)$
2:     Run QAE with oracle $\mathcal{G}_i(s, a)$ to accuracy $\varepsilon$; get $\widehat{\mu}_i$
3:     **return** $\widehat{g}_i := L + (U - L)\,\widehat{\mu}_i$
4: **end function**
5: **function** LESS$(i, j)$
6:     $\widehat{g}_i \leftarrow$ ESTIMATE$(i, \varepsilon_{\mathrm{cmp}})$;    $\widehat{g}_j \leftarrow$ ESTIMATE$(j, \varepsilon_{\mathrm{cmp}})$
7:     **return** phase $(-1)^{[\widehat{g}_i \leq \widehat{g}_j]}$ via reversible compare & uncompute
8: **end function**
9: $\widehat{i} \leftarrow$ Dürr–Høyer minimum finding over $i \in [N]$ with oracle LESS$(\cdot, \cdot)$
10: $\widehat{Q}^{\mathrm{rob}}(s, a; V) \leftarrow$ ESTIMATE$(\widehat{i}, \varepsilon_{\mathrm{est}})$
11: **return** $\widehat{Q}^{\mathrm{rob}}(s, a; V)$

---

The following proposition formalizes how much QRIM accurately estimates the robust values:

**Proposition 5** (QRIM accuracy). *For any $\varepsilon_{\mathrm{cmp}}, \varepsilon_{\mathrm{est}}, \delta \in (0, 1)$, Algorithm 2 returns $\widehat{Q}^{\mathrm{rob}}(s, a; V)$ such that, with probability at least $1 - \delta$,*

$$\left| \widehat{Q}^{\mathrm{rob}}(s, a; V) - Q^{\mathrm{rob}}(s, a; V) \right| \leq \varepsilon_{\mathrm{cmp}} + \varepsilon_{\mathrm{est}}.$$

**Remark 5.1.** *QRIM replaces the classical inner scan with a coherent minimum search. Each candidate one-step return is amplitude-encoded, pairwise comparisons are implemented via QAE at tolerance $\varepsilon_{\mathrm{cmp}}$, and Dürr–Høyer finds the minimizer; a final QAE at $\varepsilon_{\mathrm{est}}$ returns the robust backup. The linear compulation over $|\mathcal{U}|$ is reduced to $\mathcal{O}(\sqrt{|\mathcal{U}|})$ coherent queries.*

### 4.3 OUTER MAXIMIZATION INTERFACE

Value-based learners replace classical targets by

$$\mathrm{TD}^{\mathrm{rob}}(s, a, s') = r(s, a, s') + \gamma \max_{a'} \widehat{Q}^{\mathrm{rob}}(s', a'; V).$$

Policy-gradient learners form a robust critic and advantage

$$\widehat{Q}^{\mathrm{rob}}(s, a; V) \quad \text{and} \quad \widehat{A}^{\mathrm{rob}}(s, a) = \widehat{Q}^{\mathrm{rob}}(s, a; V) - V(s),$$

and use the usual estimator $\mathbb{E}[\nabla \log \pi_\theta(a|s)\, \widehat{A}^{\mathrm{rob}}(s, a)]$ with TRPO/PPO-style constraints. Rollout collection and optimization hyperparameters are unchanged. We emphasize that QRIM is compatible with both value-based or policy-gradient RL learners.

QRIM acts pointwise at $(s, a)$ and therefore realizes the rectangular robust operator:

$$(T_{\mathrm{rob}}V)(s) = \max_a Q^{\mathrm{rob}}(s, a; V)$$

For episode-wise robustness at evaluation, we fix the selected index $\widehat{i}$ for an entire trajectory and reuse the same $A_{\widehat{i}}$; the algorithm itself is unchanged.

**Remark 5.2.** *The outer learning loop is unchanged: both value-based and policy-gradient methods simply substitute the robust target $\widehat{Q}^{\mathrm{rob}}$ returned by QRIM. This makes QRIM a drop-in routine that preserves the Max–Min semantics while reducing the inner-loop cost from $\mathcal{O}(N/\varepsilon^2)$ to $\mathcal{O}(\sqrt{N}/\varepsilon)$. In essence, QRIM shows how quantum access enables robust RL to be implemented in a more scalable and practically realizable way without altering existing training pipelines.*

### 4.4 COMPUTATIONAL COMPLEXITY

We count one coherent call of query for the use of $A_i$ or $A_i^\dagger$. A classical searching with Monte Carlo uses $\mathcal{O}(N/\varepsilon^2)$ samples to achieve additive error $\varepsilon$ at $(s, a)$. QRIM achieves the quadratic speed-up beyond the classical counterpart, as formalized in the following proposition:

**Proposition 6** (Per-call query complexity). *With failure probability $\delta \in (0, 1)$, QRIM produces $\widehat{Q}^{\mathrm{rob}}(s, a; V)$ satisfying $|\widehat{Q}^{\mathrm{rob}}(s, a; V) - Q^{\mathrm{rob}}(s, a; V)| \leq \varepsilon_{\mathrm{cmp}} + \varepsilon_{\mathrm{est}}$ using $\tilde{\mathcal{O}}(\sqrt{N}\,\varepsilon_{\mathrm{cmp}}^{-1}\log(1/\delta))$ coherent queries to implement minimum finding, plus $\mathcal{O}(\varepsilon_{\mathrm{est}}^{-1}\log(1/\delta))$ for the final estimate. By choosing $\varepsilon_{\mathrm{cmp}} = \varepsilon_{\mathrm{est}} = \varepsilon/2$, QRIM requires*

$$\tilde{\mathcal{O}}\big(\sqrt{N}\,\varepsilon^{-1}\log(1/\delta)\big)$$

*total coherent queries, a quadratic improvement over $O(N\,\varepsilon^{-2})$ of the classical counterpart.*

## 5 EXPERIMENTS

We evaluate (i) **robustness** of learned policies and (ii) **inner-loop complexity** of QRIM. Training details, including RL architectures, hyperparameters, and quantum fixed-point precision, are in Appendix E. Under the assumption of access to quantum oracles for classical environment dynamics, we evaluate QRIM in two settings: a tabular domain, specifically FrozenLake-8×8, where the inner loop is implemented with simulated **coherent quantum routines**, and a continuous-control domain, specifically CartPole Swingup, where the inner loop uses a **classical emulation of a quantum circuits** while the outer learner remains classical Proximal Policy Optimization (PPO). Also, we provided extended discussions on quantum hardware applicability and report additional experimental results executed on real quantum hardware in Appendix G, demonstrating QRIM's robustness under realistic noise conditions.

### 5.1 EXPERIMENT SETUP

**Train uncertainty set vs. evaluation perturbation set.** For each environment axis, training uses an axis-aligned range discretized into a grid (the *uncertainty set*). Evaluation uses two disjoint sets per axis: *interpolation* (strictly inside the train range but excluding all train grid points) and *extrapolation* (strictly outside the train range on both sides within physical limits). We also report the nominal performance at the midpoint of the train range. The exact ranges and grid resolutions used in our implementation are summarized in Tables 1 and 2.

**Episode-wise vs. step-wise uncertainty.** Some factors are fixed for an entire episode (episode-wise), while others may change at every step (step-wise). Robust inner minimization is applied at the appropriate temporal level.

**Compared methods.** (i) **Classic RL**: non-robust training at the nominal model; (ii) **Classic RRL**: robust targets via exhaustive scan over the inner grid; (iii) **Quantum RRL** (run via QRIM): robust targets via Grover-style minimum finding (Aer emulation), keeping the outer learner unchanged.

**Protocols and metrics.** We use two protocols: *full-episodewise* (fixed-per-episode grids with repetitions) and *adversary-episodewise* (episode-wise sweep while the step-wise adversary selects worst-case at each step). We report the discounted return for both environments. Metrics include mean $\bar{J}$, $\mathrm{CVaR}_{0.1}$ (indicating lowest 10% case), and worst case over each evaluation grid $J_{\min}$; we also plot degradation curves per axis, where the nominal point serves as the reference; depending on the axis range, it appears either at the center or at the left boundary. The formulations of each metric based on the evaluation grid $\mathcal{U}_{\mathrm{eval}}$ and the return $J(\phi)$ are:

$$\overline{J} = \frac{1}{|\mathcal{U}_{\mathrm{eval}}|} \sum_{\phi} J(\phi), \qquad \mathrm{CVaR}_{0.1} = \frac{1}{\lceil 0.1|\mathcal{U}_{\mathrm{eval}}|\rceil} \sum_{i=1}^{\lceil 0.1|\mathcal{U}_{\mathrm{eval}}|\rceil} J_{(i)}, \qquad J_{\min} = \min_{\phi} J(\phi) = J_{(1)}.$$

Per-axis *degradation* relative to the train nominal is: $\Delta_k(\phi_k) = J(\phi_0) - \mathbb{E}_{\theta_{\neg k}}\big[J(\phi_k, \phi_{\neg k})\big]$.

## 5.2 ENVIRONMENT DETAILS

### FROZENLAKE-8×8 (TABULAR)

**Axes.** Episode-wise: hole layout (feasible layouts only, ordered by mean $\ell_1$ distance of holes to the grid center). Step-wise: slip probability $p \in [0, 1]$ (action executed with probability of $1 - p$, left/right neighbors with probability $p/2$ for each).

**Learner.** Tabular Q-learning variant is adopted for all methods. In robust modes, the worst hole layout is selected once per episode (episodewise), while step-wise slip probability is adversarially minimized at each transition via the inner routine (scan vs. QRIM).

Table 1: FrozenLake ranges and grids (implementation defaults). Layouts use 5 holes per case; only connected layouts are admitted.

| Factor (temporal) | Train set (range; grid) | Evaluation set | |
|---|---|---|---|
| | | Interpolation (range; grid) | Extrapolation (range; grid) |
| Hole layout (Episode-wise) | center 4×4 rim; 5 cases | center 4×4 rim; 6 cases | outer 6×6 rim; 6 cases |
| Slip probability $p$ (Step-wise) | $[0.00, 0.10]$: 4 pts | $[0.00, 0.10]$: 4 pts | $[0.10, 0.25]$: 4 pts |

### CARTPOLE SWINGUP (CONTINUOUS CONTROL)

**Axes.** Episode-wise: pole length $\ell$, mass $m$. Step-wise: slider damping $b_{\text{slider}}$, hinge damping $b_{\text{hinge}}$.

**Learner.** All methods share the same PPO actor-critic, horizons, and optimizer settings. In robust modes, the worst $(\ell, m)$ is selected once per episode (episode-wise), while step-wise damping is adversarially minimized at each transition via the inner routine (scan vs. QRIM).

Table 2: CartPole ranges and grids (implementation defaults). Interpolation excludes train points; extrapolation lies strictly outside train ranges with a 15% margin. Train grids imply $N_{\text{step}}=6\times6=36$ damping candidates and $N_{\text{episode}}=5\times5=25$ $(\ell, m)$ candidates.

| Factor (temporal) | Train set (range; grid) | Evaluation set | |
|---|---|---|---|
| | | Interpolation (range; grid) | Extrapolation (range; grid) |
| Pole length $\ell$ (Episode-wise) | $[0.50, 0.85]$; 5 pts | $[0.50, 0.85]$; 4 pts | $[0.34, 0.50], [0.85, 1.01]$; 4 pts |
| Pole mass $m$ (Episode-wise) | $[0.08, 0.22]$; 5 pts | $[0.08, 0.22]$; 4 pts | $[0.04, 0.08], [0.22, 0.28]$; 4 pts |
| Slider damping $b_{\text{slider}}$ (Step-wise) | $[0.03, 0.12]$; 6 pts | $[0.03, 0.12]$; 4 pts | $[0.02, 0.03], [0.12, 0.18]$; 4 pts |
| Hinge damping $b_{\text{hinge}}$ (Step-wise) | $[0.03, 0.12]$; 6 pts | $[0.03, 0.12]$; 4 pts | $[0.02, 0.03], [0.12, 0.18]$; 4 pts |

## 5.3 ROBUSTNESS

**FrozenLake.** In Fig. 2, we evaluate $\bar{J}$, $\text{CVaR}_{0.1}$, and $J_{\min}$ between train vs. unseen interpolation and extrapolation sets. In extrapolation (Fig. 2b), Quantum RRL mitigates performance degradation to ∼138% relative to training, significantly outperforming Classic RL which suffers a ∼204% drop, verifying superior stability against out-of-distribution scenarios. As depicted in Fig. 3, Per-factor degradation shows the slip axis and the layout axis of the hole index. The gray dashed line indicates the boundary splitting interpolation and the extrapolation sets. Quantum RRL exhibits substantial robustness, slightly better than the Classic RRL, which confirms its robustness benefits surpass those of non-robust classical RL approaches. Table 3 reports the numeric performance values.

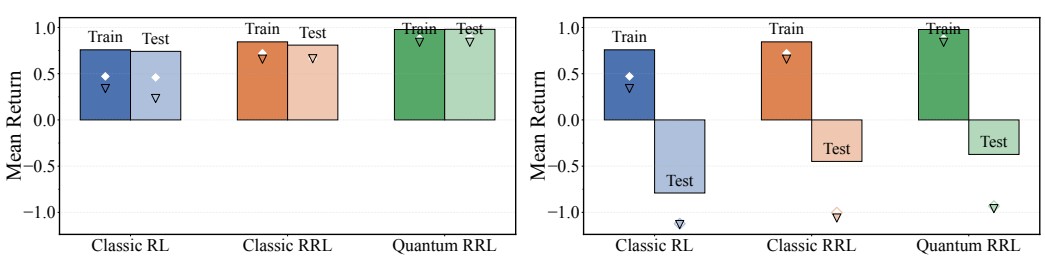

(a) Interpolation robustness evaluation  (b) Extrapolation robustness evaluation

Figure 2: FrozenLake robustness performance. Height of bar: $\bar{J}$, ⋄: $\text{CVaR}_{0.1}$, ▽: $J_{\min}$

Table 3: Robustness summary ($\bar{J}$ / CVaR$_{0.1}$ / $J_{\min}$). Left: FrozenLake, Right: CartPole Swingup.

| Method | FrozenLake | | | CartPole | | |
|---|---|---|---|---|---|---|
| | Train | interpolation | extrapolation | Train | interpolation | extrapolation |
| Classic RL | 0.76 / 0.47 / 0.34 | 0.74 / 0.46 / 0.23 | -0.79 / -1.12 / -1.13 | 61.95 / 18.04 / 11.26 | 60.66 / 21.59 / 17.84 | 26.65 / -2.94 / -3.40 |
| Classic RRL | 0.84 / 0.72 / 0.66 | 0.81 / 0.66 / 0.66 | -0.45 / -1.00 / -1.06 | 61.03 / 25.29 / 7.36 | 63.24 / 32.99 / 21.73 | 39.86 / -22.98 / -26.09 |
| **QRIM (ours)** | **0.98 / 0.88 / 0.84** | **0.98 / 0.89 / 0.84** | **-0.37 / -0.93 / -0.96** | **76.41 / 18.42 / 6.11** | **80.06 / 23.14 / 13.08** | **58.06 / -41.97 / -122.99** |

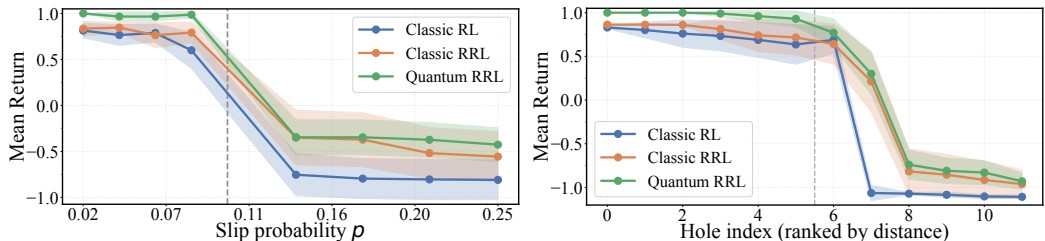

(a) Slip (step-wise) robustness evaluation (b) Hole-index (episode-wise) robustness evaluation

Figure 3: FrozenLake robustness against uncertainty factors

**CartPole.** In Fig. 4, Quantum RRL achieves superior average robustness, mitigating degradation to ~24% (against ~57% in Classic RL). It shows lower worst-case performance in extrapolation (Fig. 4b), which stems from the probabilistic nature of quantum search ($1 - \delta$ success probability). Given that Quantum RRL utilizes significantly fewer queries than Classic RRL, allocating more queries is expected to bridge this gap by increasing the minimization confidence. As illustrated in Fig. 9, Per-factor degradation isolates sensitivity along $m$, and $b_{\text{joint}}$. $\ell$ and $b_{\text{slider}}$ showed minimal variation and are provided in Appendix E. The numeric performance values are shown in Table 3.

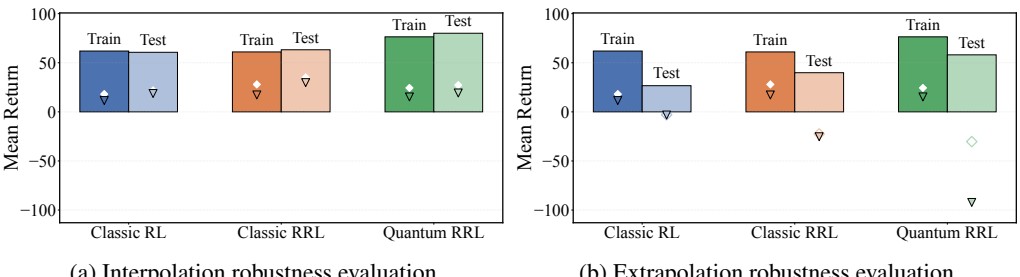

(a) Interpolation robustness evaluation (b) Extrapolation robustness evaluation

Figure 4: CartPole robustness performance. Height of bar: $\bar{J}$, $\diamond$: CVaR$_{0.1}$, $\nabla$: $J_{\min}$

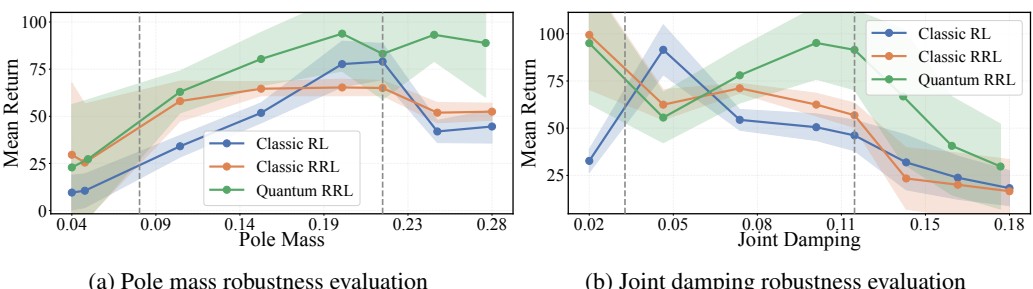

(a) Pole mass robustness evaluation (b) Joint damping robustness evaluation

Figure 5: CartPole robustness against two factors, i.e., pole mass and joint damping

## 5.4 QUERY COMPLEXITY

**Protocol.** For classical RRL we count model-evaluation calls across all $N$ candidates. For QRIM we report *median coherent oracle calls* per robust backup. Unless otherwise stated, accuracy is fixed at $\varepsilon_{\text{cmp}} = \varepsilon_{\text{est}} = 10^{-2}$ and success probability $1 - \delta = 0.9$. For the asymptotic scaling analysis (Fig. 6), we vary the uncertainty grid size $N$. Additionally, to evaluate the practical benefits during the entire training process (Fig. 7 and Table 4), we use fixed grid sizes and measure the cumulative oracle queries required for convergence.

**FrozenLake.** We vary the slip-grid size $N$; the log–log plot shows inner queries versus $N$. QRIM follows a slope $\approx \frac{1}{2}$ while classical scans follow $\approx 1$ (Fig. 6a), matching the quadratic improvement.

**CartPole.** We vary the per-step damping-grid resolution while keeping the episode-wise $(\ell, m)$ grid fixed; inner queries versus $N$ again exhibit the same slope pattern (Fig. 6b).

The advantage in query complexity translates into a substantial reduction in the total cumulative oracle calls throughout the training process. As shown in Table 4, QRIM presents a substantial reduction in query counts compared to classical RRL. Specifically, QRIM only requires 35.9% and 20.5% of the query counts of classical RRL in FrozenLake and CartPole environments, respectively. To demonstrate the speed-up of the training of QRIM, we plot the learning curves in Fig. 7. QRIM exhibits a significantly faster converging behavior, along with the number of queries spent. Notably, in the CartPole case, QRIM already reaches almost final performance within 250K queries, whereas classical RRL still struggles to learn a meaningful policy after pouring more than 200K queries.

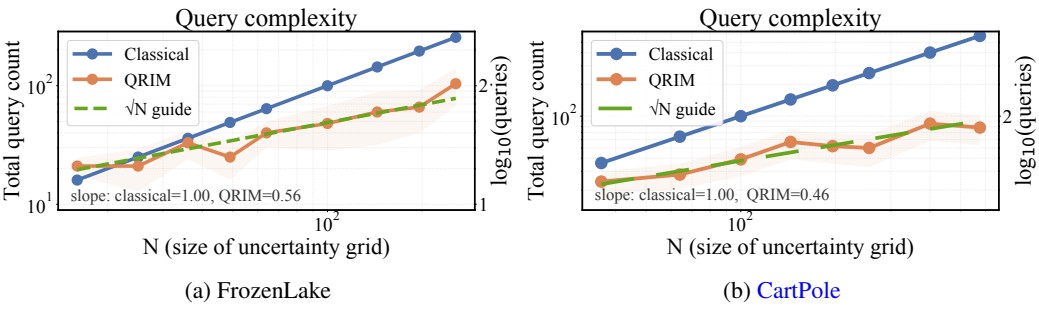

(a) FrozenLake        (b) CartPole

Figure 6: inner queries vs. $N$ (log–log). QRIM slope $\approx \frac{1}{2}$, classical $\approx 1$.

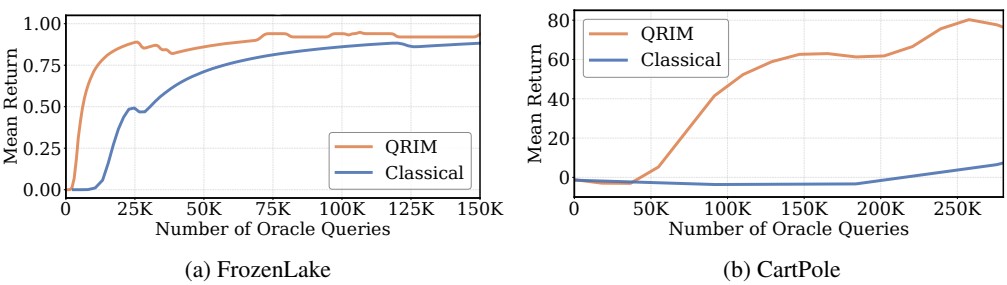

(a) FrozenLake        (b) CartPole

Figure 7: Learning curves: Mean Return vs. Number of Oracle Queries.

Table 4: Comparison of total cumulative oracle (query) calls during training.

| Environment | $N$ | Classical RRL | QRIM |
|---|---|---|---|
| FrozenLake | 4 | 11.45M | **4.11M** (35.9% of Classical RRL) |
| CartPole | 36 | 27.51M | **5.65M** (20.5% of Classical RRL) |

## 6  CONCLUSION AND LIMITATIONS

In this paper, we propose quantum robust inner minimization (QRIM), which demonstrates quadratic speed-up in query complexity beyond its classical counterpart in the robust policy training of RRL. Our approach combines a quantum perspective and the sample scalability issue of robust training for deep models. Specifically, QRIM utilizes a quantum-accessible environment to search the worst-case within the uncertain set, reducing query complexity via quantum minimum searching. In simulations via classical simulators with a coherent quantum routine and real quantum hardware, we confirm that QRIM trains a robust policy in discrete (FrozenLake) and continuous (CartPole) problems with square root reduction of quantum query calls. While we demonstrated feasibility via oracle-based encoding on NISQ hardware, a full quantum implementation for complex environments would require deep arithmetic circuits that exceed current hardware capabilities. Furthermore, the hybrid nature of our setup introduces classical-quantum communication overheads. We anticipate that practical wall-clock speedups will be fully realized as quantum hardware matures towards fault-tolerant architectures.

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

# A  APPENDIX

Proofs, experiment hyperparameters, circuit depth statistics, backend information, and extended explanations for the experiments.

# B  NOTATION FOR QUANTUM-ACCESSIBLE ROBUST RL

Table 5 summarizes the quantum notation and operators used throughout the paper.

| Symbol | Meaning |
|---|---|
| $\lvert \psi \rangle$ | Quantum state vector in Dirac notation |
| $\lvert s, a \rangle$ | Basis state encoding environment state $s$ and action $a$ |
| $\mathcal{U}$ | Uncertainty set (layouts or slips) |
| $U_P$ | Transition unitary for uncertainty index $i$, encodes $p_i(s'\lvert s, a)$ |
| $U_R$ | Reward oracle/unitary, attaches $r(s, a, s')$ to an ancilla register |
| $U_V$ | Value oracle/unitary, encodes value estimate $V(s')$ |
| $A$ | Amplitude encoding operator (prepares distribution over candidates) |
| $\mathcal{O}$ | Oracle marking "low-reward" candidates in Grover/QRIM |
| $g(s, a; V)$ | One-step backup(Bellman) target |
| $Q^{\mathrm{rob}}(s, a)$ | Robust target $\min_{p \in \mathcal{U}} g(s, a; V)$ |
| $\varepsilon_{\mathrm{cmp}}$ | Comparator tolerance in quantum min-finding |
| $\varepsilon_{\mathrm{est}}$ | Estimation tolerance (accuracy of returned robust value) |

Table 5: Quantum-related notation and operators used in our framework.

# C  PROOFS

## C.1  AMPLITUDE ENCODING OF $g_i$ (PROPOSITION 4)

*Proof.* Under Assumption 1, $x := r_i(s, a, s') + \gamma V(s') \in [L, U]$. Define the normalized quantity

$$u(s') \;:=\; \frac{x - L}{U - L} \;\in\; [0, 1]. \tag{4}$$

Consider registers $\lvert s, a \rangle \lvert 0_{S'} \rangle \lvert 0_{\mathrm{wrk}} \rangle \lvert 0_{\mathrm{anc}} \rangle$. Apply the transition sampler for candidate $i$:

$$U_P : \; \lvert s, a, 0 \rangle \;\mapsto\; \sum_{s' \in \mathcal{S}} \sqrt{P_i(s'\lvert s, a)} \, \lvert s, a, s' \rangle. \tag{5}$$

Load reward and value to a work register via reversible LUTs:

$$U_R : \; \lvert s, a, s', 0 \rangle \;\mapsto\; \lvert s, a, s', r_i(s, a, s') \rangle, \tag{6}$$

$$U_V : \; \lvert s', 0 \rangle \;\mapsto\; \lvert s', V(s') \rangle. \tag{7}$$

Compute $x = r_i(s, a, s') + \gamma V(s')$ in fixed point, then $u = (x - L)/(U - L)$ via a reversible divider. Perform a controlled $R_y$ on the ancilla with angle $2 \arcsin \sqrt{u}$:

$$\lvert u \rangle \lvert 0_{\mathrm{anc}} \rangle \;\mapsto\; \lvert u \rangle \left( \sqrt{u} \lvert 1 \rangle + \sqrt{1 - u} \lvert 0 \rangle \right). \tag{8}$$

Uncompute all work registers. The overall state becomes

$$\sum_{s'} \sqrt{P_i(s'\lvert s, a)} \, \lvert s, a, s' \rangle \, \left( \sqrt{u(s')} \lvert 1 \rangle + \sqrt{1 - u(s')} \lvert 0 \rangle \right). \tag{9}$$

Let $\mu_i := \mathbb{E}_{s' \sim P_i}[u(s')]$. By linearity of expectation, amplitude estimation on the ancilla yields $\widehat{\mu}_i$ with $\lvert \widehat{\mu}_i - \mu_i \rvert \le \varepsilon$ using $\mathcal{O}(\varepsilon^{-1} \log(1/\delta))$ coherent queries. Since

$$g_i(s, a; V) \;=\; \mathbb{E}_{s' \sim P_i}[x] \;=\; L + (U - L)\, \mu_i, \tag{10}$$

we output $\widehat{g}_i := L + (U - L)\, \widehat{\mu}_i$. □

## C.2 QRIM ACCURACY (PROPOSITION 5)

*Proof.* Let the true ordered values be $g_{(1)} \leq g_{(2)} \leq \cdots \leq g_{(N)}$ with minimizer index set $\mathcal{I}^\star := \arg\min_i g_i$. Each comparator call LESS$(i, j)$ returns the phase $(-1)^{[\widehat{g}_i \leq \widehat{g}_j]}$ where

$$|\widehat{g}_i - g_i| \leq \varepsilon_{\mathrm{cmp}}, \quad |\widehat{g}_j - g_j| \leq \varepsilon_{\mathrm{cmp}} \tag{11}$$

hold simultaneously with probability $\geq 1 - \delta_{\mathrm{cmp}}$ by QAE boosting. Thus, whenever $g_i \leq g_j - 2\varepsilon_{\mathrm{cmp}}$ the comparator returns the correct order. Equivalently, the comparator may err only on *near ties*, i.e.,

$$|g_i - g_j| \leq 2\varepsilon_{\mathrm{cmp}}. \tag{12}$$

Dürr–Høyer minimum finding with a noisy comparator of error rate $\eta < 1/2$ returns some $\hat{i}$ satisfying (see, e.g., noisy-oracle analyses)

$$g_{\hat{i}} \leq g_{(1)} + 2\varepsilon_{\mathrm{cmp}} \tag{13}$$

with probability at least $1 - \delta/2$ after polylogarithmic repetitions (absorbed into $\tilde{\mathcal{O}}(\cdot)$). Finally, we re-estimate $g_{\hat{i}}$ to accuracy $\varepsilon_{\mathrm{est}}$:

$$|\widehat{g}_{\hat{i}} - g_{\hat{i}}| \leq \varepsilon_{\mathrm{est}} \tag{14}$$

with probability at least $1 - \delta/2$. By a union bound and equation 13–equation 14,

$$\left|\widehat{g}_{\hat{i}} - g_{(1)}\right| \leq \underbrace{\left|\widehat{g}_{\hat{i}} - g_{\hat{i}}\right|}_{\leq \varepsilon_{\mathrm{est}}} + \underbrace{\left|g_{\hat{i}} - g_{(1)}\right|}_{\leq 2\varepsilon_{\mathrm{cmp}}} \leq 2\varepsilon_{\mathrm{cmp}} + \varepsilon_{\mathrm{est}} \tag{15}$$

with probability at least $1 - \delta$. Replacing the factor 2 by 1 is possible by setting the decision threshold in LESS at $\varepsilon_{\mathrm{cmp}}$-margins; we keep the simpler form and state the bound as $\varepsilon_{\mathrm{cmp}} + \varepsilon_{\mathrm{est}}$ up to a constant factor absorbed by redefining $\varepsilon_{\mathrm{cmp}}$ (customarily done in QAE analyses). □

## C.3 PER-CALL QUERY COMPLEXITY (PROPOSITION 6)

*Proof.* Throughout this section, we use $M_{\mathrm{alg}}(\cdot)$ to denote the *query complexity* (i.e., the number of coherent oracle calls) required by algorithm "alg" at the specified accuracy and confidence parameters. Here $A_i$ denotes the amplitude-preparation unitary for candidate $i$, which maps the all-zero state to the superposition in equation 9, and $A_i^\dagger$ is its inverse. A single QAE call achieving additive tolerance $\varepsilon$ with failure $\leq \delta'$ uses

$$M_{\mathrm{QAE}}(\varepsilon, \delta') = \mathcal{O}\big(\varepsilon^{-1} \log(1/\delta')\big) \tag{16}$$

coherent queries to $A_i$ and $A_i^\dagger$ (Brassard et al., 2002). The Dürr–Høyer search over $N$ items with access to a *mark* oracle (here, a comparator phase) has complexity

$$M_{\mathrm{DH}}(N, \delta/2) = \tilde{\mathcal{O}}\big(\sqrt{N} \log(1/\delta)\big). \tag{17}$$

Implementing one comparator requires *two* QAE calls at accuracy $\varepsilon_{\mathrm{cmp}}$; hence the search phase costs

$$M_{\mathrm{search}} = \tilde{\mathcal{O}}\big(\sqrt{N} \cdot M_{\mathrm{QAE}}(\varepsilon_{\mathrm{cmp}}, \delta')\big) = \tilde{\mathcal{O}}\big(\sqrt{N} \, \varepsilon_{\mathrm{cmp}}^{-1} \log(1/\delta)\big), \tag{18}$$

where we set $\delta' = \Theta(\delta/\log N)$ and absorb polylog factors. The final estimate contributes

$$M_{\mathrm{final}} = \mathcal{O}\big(\varepsilon_{\mathrm{est}}^{-1} \log(1/\delta)\big). \tag{19}$$

Choosing $\varepsilon_{\mathrm{cmp}} = \varepsilon_{\mathrm{est}} = \varepsilon/2$ yields the total

$$M_{\mathrm{QRIM}} = \tilde{\mathcal{O}}\big(\sqrt{N} \, \varepsilon^{-1} \log(1/\delta)\big). \tag{20}$$

A classical exhaustive scan that estimates every $g_i$ by Monte Carlo to accuracy $\varepsilon$ requires

$$M_{\mathrm{classical}} = \Theta\big(N \, \varepsilon^{-2} \log(1/\delta)\big) \tag{21}$$

samples, giving the quadratic improvement simultaneously in $N$ and $1/\varepsilon$. □

## C.4 APPROXIMATE ROBUST OPERATOR AND FIXED-POINT STABILITY

Let $\widehat{Q}^{\mathrm{rob}}(s, a; V)$ satisfy the uniform bound

$$\left| \widehat{Q}^{\mathrm{rob}}(s, a; V) - Q^{\mathrm{rob}}(s, a; V) \right| \leq \epsilon, \qquad \forall(s, a), \ \forall V, \tag{22}$$

with $\epsilon := \varepsilon_{\mathrm{cmp}} + \varepsilon_{\mathrm{est}} + c\eta_{\mathrm{arith}}$. Define

$$(\widehat{T}_{\mathrm{rob}}V)(s) := \max_a \widehat{Q}^{\mathrm{rob}}(s, a; V). \tag{23}$$

**Lemma 1** (Operator deviation). *For any $V$, $\|(\widehat{T}_{\mathrm{rob}} - T_{\mathrm{rob}})V\|_\infty \leq \epsilon$.*

*Proof.* For each $s$,

$$\left| (\widehat{T}_{\mathrm{rob}}V)(s) - (T_{\mathrm{rob}}V)(s) \right| = \left| \max_a \widehat{Q}^{\mathrm{rob}}(s, a; V) - \max_a Q^{\mathrm{rob}}(s, a; V) \right| \tag{24}$$

$$\leq \max_a \left| \widehat{Q}^{\mathrm{rob}}(s, a; V) - Q^{\mathrm{rob}}(s, a; V) \right| \leq \epsilon. \tag{25}$$

$\square$

**Theorem 1** (Fixed-point perturbation). *Let $V^*$ be the unique fixed point of $T_{\mathrm{rob}}$ and $\widehat{V}$ be the unique fixed point of $\widehat{T}_{\mathrm{rob}}$. Then*

$$\| \widehat{V} - V^* \|_\infty \leq \frac{\epsilon}{1 - \gamma}. \tag{26}$$

*Proof.* By $\gamma$-contraction of $T_{\mathrm{rob}}$ and $\widehat{T}_{\mathrm{rob}}$ and Lemma 1,

$$\| \widehat{V} - V^* \|_\infty = \| \widehat{T}_{\mathrm{rob}}\widehat{V} - T_{\mathrm{rob}}V^* \|_\infty \tag{27}$$

$$\leq \| \widehat{T}_{\mathrm{rob}}\widehat{V} - T_{\mathrm{rob}}\widehat{V} \|_\infty + \| T_{\mathrm{rob}}\widehat{V} - T_{\mathrm{rob}}V^* \|_\infty \tag{28}$$

$$\leq \epsilon + \gamma\| \widehat{V} - V^* \|_\infty. \tag{29}$$

Rearrange to obtain equation 26. $\square$

# D   DETAILED CONSTRUCTION OF QUANTUM ENVIRONMENT ORACLES

This section provides the conceptual circuit-level design of the full quantum oracle unitaries $(U_P, U_R, U_V)$ defined in Definition 2. These constructions serve to demonstrate the structural feasibility of our quantum-accessible environment framework and clarify the explicit structure required for a fully coherent quantum oracles. In our experiments, we assume noise-free quantum oracles without explicitly constructing quantum circuits. We present two representative environments that serve complementary roles in demonstrating conceptual feasibility and practical scalability.

## D.1   FROZENLAKE: COHERENT LOGIC CIRCUITS

For the discrete state space, we construct the full quantum oracles as explicit quantum circuits using standard quantum logic primitives. The process creates a coherent data flow through three stages without any classical intervention during the quantum routine.

**Transition Probability Encoding** ($U_P$)   The transition oracle constructs the next-state superposition by combining grid geometry logic with stochastic mixing. The circuit begins with the state $|s\rangle$ and action $|a\rangle$ registers. To encode stochasticity, a multi-controlled $R_y(\theta)$ gate (with $\theta = 2\arcsin(\sqrt{p})$) acts on an auxiliary slip-flag qubit, conditioned on the action register. This creates a superposition of intended ($|0\rangle_{\mathrm{flag}}$) and slip ($|1\rangle_{\mathrm{flag}}$) branches. Controlled on these flag states, logical arithmetic gates update the state register $|s\rangle$ to compute the intended next state (e.g., adding the grid width to the index for a Down move) or the slip state. This results in the entangled next-state register $|\psi\rangle_{next} = \sqrt{1-p}|s_{\mathrm{intended}}\rangle + \sqrt{p}|s_{\mathrm{slip}}\rangle$.

**Reversible Value Loading** ($U_R, U_V$)   The Reward Oracle $U_R$ realizes the reward function via boolean logic. We implement this using a network of Multi-Controlled X (MCX) gates. For instance, if a specific next-state index $s'$ corresponds to a Hole (reward $-1$), an MCX gate targeting the reward register $|r\rangle$ is activated only when the state register matches the binary string of that cell's index. Similarly, the Value Oracle $U_V$ loads the estimated value $V(s')$ via a QROM lookup structure. Finally, a reversible Quantum Adder computes the total return $G = r + \gamma V$, which drives a controlled rotation to encode the amplitude. Crucially, all intermediate compute logic is uncomputed to disentangle the registers.

**Amplitude Encoding**   Finally, to realize the quantum minimum finding for the objective $f(i) = g_i(s, a; V)$ (as defined in Section 3.3), we combine the loaded values. A reversible Quantum Adder first computes the raw return $G = r + \gamma V$. Following Assumption 1, this value is normalized to $u = (G - L)/(U - L)$. This scalar $u$ drives a controlled-$R_y(\theta)$ rotation (with $\theta = 2\arcsin(\sqrt{u})$) on a single ancilla qubit. This step strictly implements the state-preparation unitary $\mathcal{G}_i(s, a)$ described in Proposition 4, encoding the robust objective into the amplitude $\sqrt{u}\,|1\rangle$. Crucially, after this encoding, all intermediate compute logic ($U_R, U_V$, Adder) is uncomputed to disentangle the registers.

### D.2   CARTPOLE: CLASSICAL EMULATION OF QUANTUM ORACLES

For the CartPole environment, a full circuit implementation requires encoding complex differential equations (e.g., friction, contact dynamics) into reversible quantum arithmetic circuits. As such deep circuits exceed the coherence limits of current hardware and simulators, we employ a classical emulation strategy that mathematically simulates the action of ideal quantum oracles.

**Transition Probability Encoding** ($U_P$)   A rigorous quantum circuit for CartPole dynamics ($U_P$) would necessitate fixed-point quantum registers representing continuous variables $(x, \dot{x}, \theta, \dot{\theta})$ and a sequence of reversible adders, multipliers, and CORDIC circuits to perform time-stepping integration. Since simulating these substantial number of gates is computationally prohibitive and prone to noise on current devices, we instead calculate the resulting quantum amplitudes classically. Our emulation reconstructs the exact quantum state vector that an ideal circuit would produce. In the classical computation step, for a given superposition of state-action pairs $\sum \alpha_i\,|s_i, a_i\rangle$, the emulator queries the classical physics engine (MuJoCo) to compute the deterministic next state $s_i' = f_{\text{physics}}(s_i, a_i)$ for each basis state $i$, effectively reconstructing the exact quantum state vector.

**Reversible Value Loading** ($U_R, U_V$)   Theoretically, the reward ($U_R$) and value ($U_V$) oracles require arithmetic circuits to evaluate the analytic reward equation and the value network reversibly. In our emulation, we bypass the gate-level construction and directly compute the reward $r_i$ and value estimate $V(s_i')$ using classic arithmetic based on the state $s_i'$ obtained from the transition step. This process mimics the coherent loading of data into quantum registers $|r, V\rangle$ without the overhead of simulating deep arithmetic logic, ensuring that the values required for the robust objective $g_i(s, a; V)$ are correctly associated with each superposition branch.

**Amplitude Encoding**   Finally, to enable Quantum Amplitude Estimation, the loaded values must be encoded into the amplitude of an ancilla qubit. While a full circuit would employ a reversible Quantum Adder and a controlled rotation, our emulator performs Amplitude Injection. We mathematically calculate the normalized return $u_i = (r_i + \gamma V(s_i') - L)/(U - L)$ following Assumption 1, and directly inject the corresponding amplitude into the state vector: $|\Psi\rangle = \sum_i \alpha_i\,|s_i, a_i\rangle\,(\sqrt{u_i}\,|1\rangle_{\text{anc}} + \dots)$. This constructed state is then passed to the Grover diffusion operator. By counting this batch computation as a single coherent oracle call, we validate the algorithmic query complexity ($\mathcal{O}(\sqrt{N})$) of QRIM, isolating the quantum advantage from hardware constraints.

## E   EXPERIMENTS DETAILS

**FrozenLake (tabular, value-based).**   We use the $8\times8$ map with start $(0, 0)$ and goal $(7, 7)$. Episode-wise uncertainty is given by hole layouts sampled from the central $4\times4$ rim, with $H{=}5$ holes per layout. Six layouts are used for training; interpolation layouts come from the same rim but disjoint from training; extrapolation layouts come from the $6\times6$ rim. Step-wise uncertainty is

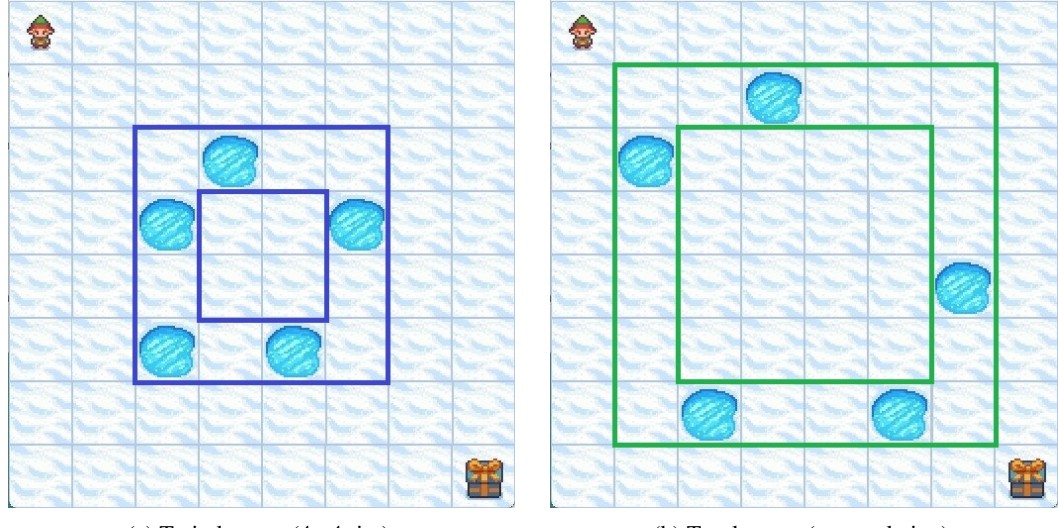

(a) Train layouts (4×4 rim).

(b) Test layouts (extrapolation).

Figure 8: **FrozenLake uncertainty sets.** Left: training layouts sampled on the central 4×4 rim. Right: evaluation layouts for extrapolation (central 6×6 rim).

the slip probability $p$, with training grid $\{0.000, 0.033, 0.067, 0.100\}$; interpolation uses midpoints inside $(0.000, 0.100)$; extrapolation extends up to $0.25$. Tabular Q-learning is used with $\gamma=0.99$, learning rate $\alpha=0.1$, and $\epsilon$-greedy exploration decayed linearly from $0.10$ to $0.02$ over the first 30% of episodes. Exploring starts are used with probability $0.35$. Robust variants blend worst-case and mean backups with factor $0.7$, and select the current worst layout with probability $0.75$.

**CartPole swingup (dm_control, policy-gradient).** Episode-wise uncertainty is the pole length $\ell$ and mass $m$; step-wise uncertainty is slider damping $b_{\text{slider}}$ and joint damping $b_{\text{joint}}$. Training grids: $5 \times 5$ for $(\ell, m)$ and $6 \times 6$ for $(b_{\text{slider}}, b_{\text{joint}})$; interpolation excludes training points; extrapolation expands ranges by 15%. We train PPO with identical hyperparameters across baselines: two-layer MLP (128 units, Tanh) for actor and critic, clip ratio $0.2$, learning rates $3 \times 10^{-4}$ (policy) and $10^{-3}$ (value), GAE $\lambda=0.95$, horizon 2048, minibatch size 256, and 40 SGD updates per epoch. Discount $\gamma=0.99$. Robust RRL scans all damping candidates per step; QRIM calls Grover min-finding on the damping grid while leaving the PPO pipeline unchanged. Episode-wise robust choice of $(\ell, m)$ is made using the critic at the start of each rollout.

# F ADDITIONAL CARTPOLE DEGRADATION PLOTS

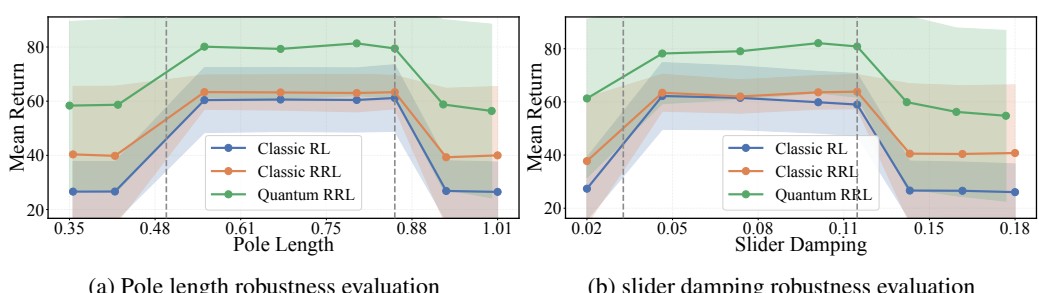

(a) Pole length robustness evaluation

(b) slider damping robustness evaluation

Figure 9: CartPole robustness against additional factors, i.e., pole length and slider damping

# G  DISCUSSIONS: QUANTUM HARDWARE VERIFICATION

To address the concerns regarding the feasibility of our approach on physical quantum devices and to validate the noise resilience of QRIM beyond simulations, we conducted additional experiments using both a realistic noise simulator and real quantum hardware. For a noise model simulator, we tested with Qiskit Aer noise model, applying depolarizing error rates of 0.05% and 0.6% for 1-Qubit and 2-Qubit error respectively, and a 2% readout error. These values were selected to sufficiently reflect the limits and operational error rates of contemporary quantum devices. For real quantum hardware, we used IBM Quantum backend (ibm-anonymous institute) with 127-qubit Eagle processor, with 2 qubits and 20 depth circuits, isolating the quantum search routine while assuming access to quantum oracles for environment dynamics.

## G.1  ROBUSTNESS TO QUANTUM NOISE

We evaluated the robustness of QRIM across three conditions: noise-free simulation, noise model simulation, and real quantum hardware execution. All experiments were conducted on the FrozenLake-$8\times8$ environment under the extrapolation setting, consistent with the setup described in Section 5. The results are summarized in Table 6.

Table 6: Comparison of QRIM performance on real quantum hardware.

| Metric | FrozenLake | | |
| --- | --- | --- | --- |
| | **QRIM noise-free** | **QRIM noise model** | **QRIM quantum HW** |
| Train | 0.98 | 0.91 | 0.94 |
| Test (extrapolation) | $-0.37$ | $-0.44$ | $-0.52$ |

As shown in the Table 6, QRIM executed on real quantum hardware achieved an extrapolation return of -0.52, which is slightly lower than the Classic RRL (-0.45) and the noise-free QRIM (-0.37) due to hardware noise, but significantly outperforms Classic RL(-0.79).

The results confirm that QRIM maintains the performance trend observed in the noise model simulations (-0.44), demonstrating practical robustness even on noisy intermediate-scale quantum devices (NISQ). Also, the quadratic query speed-up ($\mathcal{O}(\sqrt{\mathcal{U}})$) was preserved on real quantum hardware. The total number of queries had been increased only for 5% compared to the noise free QRIM, which is still much fewer queries than classical RRL.

## G.2  DISCUSSIONS ON QRIM'S IMPLEMENTATION ON QUANTUM HARDWARE

In addition, our theory provides an intuition about its practicality in noisy quantum hardware. In the scale behavior of QRIM, i.e., $\tilde{\mathcal{O}}(\sqrt{N}\log(1/\delta))$ (referring to Proposition 3), $\delta$ means the failure probability for minimum search. In our noise-free simulations, we already set an imperfect search with a success probability of $(1-\delta) = 0.90 < 1$. It reflects the stochastic behavior of quantum environments. In the related literature, hardware noise is shown to degrade the success probability, which is another factor that might affect the success of minimum search.

As we already assume imperfect minimum search, i.e., $(1-\delta) = 0.90$, in our experiments, QRIM's successful training even with such imperfect search makes us conjecture QRIM's possible robustness against hardware noise, which also disturbs perfect search.

This leads us to conjecture that QRIM shows sufficient robustness in both noise-model-based experiments and real quantum hardware, coinciding with the previous noise-free results. Also, if we set a severe failure probability (a high $\delta$) for QRIM, it may lead to a slight increase in the required query counts to keep the robustness performance.

