# OpenReview forum: "QRIM: Quantum Robust Inner Minimization for Reinforcement Learning"
_ICLR.cc/2026/Conference — ICLR 2026 Conference Desk Rejected Submission_

### Official Review · Reviewer_YZjp · 2025-10-21

**Soundness:** 3
**Presentation:** 3
**Contribution:** 3
**Rating:** 8
**Confidence:** 4

**Summary:**

This work deals with the potential benefits of quantum computing for RL. It demonstrates that in the case of robust RL, where worst-case performance is optimized across a set of potential environments, quantum techniques can achieve a quadratic advantage in sample efficiency.

**Strengths:**

I find it particularly strong that the work highlights another area in which QC can offer an advantage for RL, especially since it is a practice-relevant area (robust RL).

**Weaknesses:**

I see no real weakness, but the integration with current similar work could be a bit stronger. In particular, an outlook or assessment of further development and the use of real quantum hardware would be beneficial.
For example, [1], in which the algorithm from [2], which, like the present work, demonstrates a quadratic advantage in sample efficiency for RL, was investigated in minimal size on real quantum hardware.


Further:
"kernal" -> "kernel"
"markov" -> Markov" (use {Markov})
"wasserstein" -> "Wasserstein"
"bellman" -> "Bellman"


[1] D. Hein et al., From Classical Data to Quantum Advantage--Quantum Policy Evaluation on Quantum Hardware, 2025

[2] S. Wiedemann et al., Quantum Policy Iteration via Amplitude Estimation and Grover Search – Towards Quantum Advantage for Reinforcement Learning, 2023

**Questions:**

* Would you rather place your work in online RL or offline RL?
* Would mentioning the QRL survey [3] be useful, or does that not fit with the chosen grouping of QRL areas?

[3] N. Meyer et al., A survey on quantum reinforcement learning, 2022

---

> ### Author Response · Authors · 2025-11-20
> **Response to Reviewer YZjp (Part 1/1)**
>
> ### **Dear Reviewer YZjp**,
> We sincerely appreciate your constructive comments. We hope that our response relieves your concerns.
>
> &nbsp;
>
> >### **Weakness 1. Outlook or assessment of further development and the use of real quantum hardware would be beneficial**
>
> Quantum hardware validation is a direct way to demonstrate the scalability and practicality of the idea. We acknowledge that this is particularly important in the quantum world.
>
> However, our initial goal was to **theoretically** figure out how a quantum nature brings quadratic query speed-up to robust deep learning.
>
> We have tried our best to extend our theory to quantum hardware in the rebuttal.
>
> We are indeed conducting a small-scale **demonstration** on ‘**real’ quantum hardware** for the FrozenLake environment, thanks to the unexpected quantum resources we can now temporarily use. It is not guaranteed, but we sincerely hope to report the results within the rebuttal period. Also, we ask for your kind understanding that the quantum hardware is still NOT prevalent, and the current commercialized cloud services are extremely costly (hard to support with our research funding).
>
> **Noise modelling:** **We additionally tested QRIM with noise modelling, i.e., Qiskit Aer noise model** (with depolarizing error rates of 0.05% and 0.6% for 1-Qubit and 2-Qubit error, respectively, and a 2% readout error).
>
> We found that QRIM consistently shows its superiority even in the noise model with minimal changes in performance, with only 4% increase of total query counts. (still much fewer queries and better robustness than classical RRL).
>
> We have added this result to the revised manuscript (Appendix G).
>
> |  | **QRIM noise-free** | **QRIM w/ noise** |
> | --- | --- | --- |
> |  | FrozenLake | FrozenLake |
> | train | **0.98** | 0.91 |
> | test | **-0.37** | -0.44 |
>
> &nbsp;
>
> >### **Weakness 2. Minor errors**
>
> Thank you for the careful proofreading. We have corrected all the listed typos:
> - In L164, 166: Corrected “kernal” to “kernel”
> - In References i) Corrected “markov” to “Markov”, ii) Corrected “wasserstein” to “Wasserstein”, iii) Corrected “bellman” to “Bellman”
>
> &nbsp;
>
> >### **Question 1. Would you rather place your work in online RL or offline RL?**
>
> We place our work in **Online RL**.
>
> Our proposed algorithm (QRIM) is designed for agents that actively interact with the environment (the quantum oracle) to collect trajectories and update policies iteratively during training.
>
> This contrasts with Offline RL, which learns from a fixed, pre-collected dataset.
>
> While we utilize a "batch" of uncertainty candidates in the inner loop, the outer loop remains a standard online interaction process (e.g., PPO).
>
> &nbsp;
>
> >### **Question 2. Would mentioning the QRL survey [3] be useful, or does that not fit with the chosen grouping of QRL areas?**
>
> We agree that referencing Meyer et al. (2022) [3] is highly beneficial. We have incorporated this survey into Section 2.
>
> It provides a taxonomy that helps position our "Quantum accessible environment" approach within the broader landscape of QRL, distinguishing it from VQC-based policy search methods.

---

> ### Author Response · Authors · 2025-11-26
>
> ### **Dear Reviewer YZjp**,
>
> Motivated by your feedback, we have successfully conducted experiments on a real quantum hardware of 127-qubit IBM Eagle processor.
> Please refer to our general response(Additional Results: Real Quantum Hardware Verification) and the newly added Appendix G for the comprehensive results.

---

### Official Review · Reviewer_YhQS · 2025-10-30

**Soundness:** 3
**Presentation:** 2
**Contribution:** 3
**Rating:** 4
**Confidence:** 3

**Summary:**

The paper addresses the challenge of the high computational cost associated with Max-Min optimization in robust reinforcement learning (RRL). The authors investigate whether it is possible to accelerate the Max-Min optimization process by leveraging a quantum oracle for the environment. The paper demonstrates that the proposed approach achieves a quadratic speed-up in query complexity compared to classical RRL methods.

**Strengths:**

The manuscript is very well-written, structured, and easy to follow. The authors present QRL and RRL concepts in a clear and engaging manner. Algorithms 1 and 2 are particularly effective in bridging various components of the proposed approach. The experimental results obtained on the two benchmark environments are impressive and substantiate the claims made.

**Weaknesses:**

However, the paper lacks a detailed explanation of how the quantum environment oracles are constructed for the benchmark environments, FrozenLake and CartPole. Given that the quantum-accessible environments form the core of the proposed approach, more comprehensive information on this aspect is essential. In this context, the authors could consider referencing the following research, which demonstrates a practical implementation of a quantum oracle in reinforcement learning:

Simon Wiedemann, Daniel Hein, Steffen Udluft, and Christian B. Mendl. Quantum policy iteration via amplitude estimation and Grover search – Towards quantum advantage for reinforcement learning. Transactions on Machine Learning Research, 2023. ISSN 2835-8856.
Minor Issues:
Algorithm 2: The usage of epsilon ($\epsilon$) lacks a subscript, which should be added for clarity.
Page 6: The final sentence on this page includes “(dm control CartPole Swingup),” which could be rephrased for better clarity and to avoid potential confusion.
Page 9: There is a typographical error in Fig. 5, where “Cartplole” should be corrected to “CartPole.”

**Questions:**

How are the quantum oracles of the benchmark environment constructed?

---

> ### Author Response · Authors · 2025-11-20
> **Response to Reviewer YhQS (Part 1/1)**
>
> ### **Dear Reviewer YhQS**,
> We truly appreciate your detailed comments to revise our paper. We hope to address your concerns through the following response.
>
> &nbsp;
>
> >### **Weakness 1, Question 1: Lacks a detailed explanation of how the quantum environment oracles are constructed for the benchmark environment, How are the quantum oracles of the benchmark environment constructed?**
>
> We have added “Appendix D. Detailed Construction of Quantum Environment Oracles”  which provides:
> - Specification of the oracles in quantum accessible environment
> - Circuit descriptions for both FrozenLake and CartPole cases
>
> In summary, the quantum oracles are constructed based on three core principles:
> - **Transition Probability Encoding:** The transition sampler $U_{P}$ is implemented as a state preparation unitary that coherently encodes the transition probabilities $\sqrt{P(s'|s, a)}$ into quantum amplitudes.
> - **Reversible Value Loading:** The reward ($U_R$) and value ($U_V$) oracles utilize standard **reversible logic** to coherently load the one-step reward $r$ and value estimate $V$ onto auxiliary registers.
> - **Amplitude Encoding:** These loaded values are combined using reversible arithmetic, and a final controlled rotation is used to encode the normalized one-step return ($r+\gamma V$) into the amplitude of an ancilla qubit, preparing it for Quantum Amplitude Estimation (QAE).
>
> The inclusion of this detailed subsection significantly enhances the technical transparency and reproducibility of the quantum routine by providing explicit implementation steps for the quantum oracles.
>
> &nbsp;
>
> >### **Weakness 2: Minor Issues**
>
> Thank you for this detailed feedback. We corrected the issues as below:
> - The use of $\epsilon$ in Algorithm 2.
> - $\epsilon$ in Line 1,2 in Algorithm 2 is simply the **function definition**, where $\epsilon$ serves as a generic formal parameter for the target accuracy. as in Proposition 2. It is instantiated as $\epsilon_{cmp}$ (for comparison) in Line 6 and $\epsilon_{est}$ (for estimation) in Line 10.
> - Page 7 “dm control Cartpole Swingup”
>   - We have rephrased the sentence to provide a balanced description of both experimental settings
> > We evaluate QRIM in two settings: a tabular domain, ***specifically FrozenLake-8×8,*** and a continuous-control domain, ***specifically CartPole Swingup***.
> - Page 10 “Cartplole”: Corrected to “CartPole”

---

### Official Review · Reviewer_29uc · 2025-10-31

**Soundness:** 4
**Presentation:** 4
**Contribution:** 4
**Rating:** 8
**Confidence:** 3

**Summary:**

This paper introduces a quantum based approach to solving the inner minimization problem of robust reinforcement learning algorithms, that promises . The method assumes a quantum-accessible environment that enables the use of quantum minimum finding to speed up searching the uncertainty set for parameters that minimize the resulting reward. Two simple environments are used for evaluation: one with discrete, and one with continuous actions. The results show quadratic speed up in number of oracle calls.

**Strengths:**

The method is well motivated and preliminaries are introduced concisely. Assumptions are also clearly delineated. The computational complexity in terms of the number of queries, error and failure probability is given with proof. Experimental setup is chosen adequately to show the promised improvement empirically.

Clever use of quantum entanglement results in a tangible quantum advantage. The most convincing strength of this paper is the possibility of drop-in replacement into any existing robust RL algorithm (given the assumptions hold).

**Weaknesses:**

The method assumes a quantum-accessible environment model which is rather restrictive.
The number of environments used for evaluation is limited. Using more environments with larger action spaces would further improve the quality of the paper.

Minor errors:
- Section 3.2: "kernal"
- Section 3.3: "We stand on" worded unnaturally?
- Section 5.1: "Robust inner minimization is applied at the appropriate temporal... " missing a word?
- Section 6: "minimin"

**Questions:**

- What is the symbole $\Delta_{S}$ in Definition 1?
- Can you explain in more detail what the "aux" qbits are used for?
- In the CartPole environment, the worst-case performance of the proposed method is significantly worse than for the baseline models. Can you identify a reason for this gap?
- You are reporting **median** coherent oracle calls to assess the query complexity. Is the variance of the query count large?

---

> ### Author Response · Authors · 2025-11-20
> **Response to Reviewer 29uc (Part 1/2)**
>
> ### **Dear Reviewer 29uc,**
>
> We sincerely appreciate your thorough review. We hope to address your concerns by providing the following author response.
>
> &nbsp;
>
> >### **Weakness 1,2. A restrictive setting and limited environments with small action spaces**
>
> We acknowledge that our QRIM is applied only to environments that are successfully modeled by quantum mechanics. We all know that such modeling is challenging.
>
> However, we hope to point out that **a quantum-accessible environment offers non-trivial advantages for robust training, which is another reason we push forward with establishing quantum-accessible environments.** We emphasize that this benefit directly tackles the scalability issue of modern AI, which suffers from extensive queries to be spent.
>
> While the evaluations on environments with larger action spaces are desirable, we hope to first present the robustness benefits of quantum-accessible environments from the tractable benchmarks. (**In fact, larger action spaces require an excessive number of qubits, which are far impractical at this moment**)
>
> To give a rationale why we select FrozenLake and CartPole, we want to pick a *discrete* (state), *tabular* (Q function) domain for FrozenLake (an easier one) and, in contrast, a *continuous* (state), *function-approximated* (PPO) domain for CartPole (a harder one), covering the variants of environmental states and ways of value estimation.
>
> Another practical reason for benchmark selection is that FrozenLake (or with a similar level of complexity) is the most complicated environment with which can be tested with a full quantum routine. To the best of our knowledge, a full quantum routine evaluation of a more complex environment is not yet popular in the related literature. On the other hand, for CartPole, which is a more complicated one, we demonstrate via classical emulation tools (not a full quantum routine).
>
> Based on such a rationale, we argue that our theoretical claim is demonstrated as possibly sufficient as we can validate.
>
> &nbsp;
>
> >### **Weakness 3. Minor errors**
>
> Thank you for this detailed feedback. We corrected all of them as below (see the revised manuscript):
> - Section 3.2 L164, 166: Corrected “kernal” to “kernel”
> - Section 3.3 L192: “We stand on” to "We build upon"
> - Section 5.1 L357: added a word “level” to become "appropriate temporal level"
> - Section 6 L537: Corrected “minimim” to “minimum”
>
> &nbsp;
>
> >### **Question 1: What is the symbole $\Delta_S$ in Definition 1?**
>
> The symbol $\Delta_{\mathcal{S}}$ represents **"the set of all probability measures over the state space S."**.
>
> We have defined this in the RL Background (Line 143 in Section 3.1), but for better clarity, we have restated it in Definition 1.
>
> &nbsp;
>
> >### **Question 2. Can you explain in more detail what the "aux" qbits are used for?**
>
> The "aux" qubit is the **auxiliary (or ancilla) qubit** used in **Proposition 4**  for amplitude encoding.
>
> Its purpose is to hold the normalized 1-step return value. The state is prepared as $\sqrt{1-u}|0\rangle_{aux} + \sqrt{u}|1\rangle_{aux}$, where $u$ is the normalized 1-step return ($r + \gamma V(s')$). Our algorithm then uses Quantum Amplitude Estimation (QAE) to estimate the probability of measuring this 'aux' qubit in the $|1\rangle$ state, which allows us to efficiently estimate the expected return $g_i$.
>
> We have added details in a new subsection: **“Appendix D. Detailed Construction of Quantum Environment Oracles”**)

---

> ### Author Response · Authors · 2025-11-20
> **Response to Reviewer 29uc (Part 2/2)**
>
> >### **Question 3. The worst-case performance of the proposed method is significantly worse than for the baseline models. Can you identify a reason for this gap?**
>
> The "significantly worse" $J_{min}$ for QRIM in the CartPole extrapolation setting is not a simulation error, but a direct consequence of the probabilistic nature of our quantum algorithm; quantum measures probabilistic observations rather than deterministic exhaustive searches. Specifically, QRIM finds the worst case with a success probability $1-\delta< 1$.
>
> The Min-case is worse, but we emphasize that its mean performance is much better than the classical counterparts.
>
> Notably, QRIM can achieve a quadratic reduction of query complexity due to its quantum-based probabilistic behavior, i.e., $\mathcal{O}(\sqrt{|\mathcal{U}|}$).
>
> &nbsp;
>
>
> >### **Question 4. You are reporting median coherent oracle calls to assess the query complexity. Is the variance of the query count large?**
>
> The shaded area in Fig. 6 does indeed illustrate this variance. In fact, due to the stochasticity of quantum nature, query counts are slightly varying, but they generally follow our theoretical scaling-up curve, consistent with our theory.
>
> We report the actual standard deviations for the curves in Fig. 6:
>
> | FrozenLake | N (x-axis) | 16 | 25 | 36 | 49 | 64 | 100 | 144 | 196 | 256 |
> | --- | --- | --- | --- | --- | --- | --- | --- | --- | --- | --- |
> |  | QRIM query (y-axis) | 21 | 21 | 33 | 25 | 40 | 48 | 60 | 66 | 104 |
> |  | std of query | 2.7 | 8.2 | 9.1 | 8.9 | 6.9 | 19.6 | 24.1 | 21.2 | 25.8 |
>
> | Cartpole | N (x-axis) | 36 | 64 | 100 | 144 | 196 | 256 | 400 | 576 |
> | --- | --- | --- | --- | --- | --- | --- | --- | --- | --- |
> |  | QRIM query (y-axis) | 24 | 28 | 39 | 57 | 52 | 50 | 85 | 78 |
> |  | std of query | 3.0 | 7.4 | 13.8 | 15.7 | 16.4 | 15.3 | 20.0 | 25.5 |
>
> **We hope to emphasize that QRIM’s query counts benefits are still solid, even considering the standard deviation.** For example, for FrozenLake, N=256 leads to the excessive query counts, i.e., $256$, for classical RRL, but QRIM shows merely $104\pm25.8$ counts. Also, the CartPole case shows $576$ counts are required for classical RRL, but QRIM needs $78\pm25.5$ counts.
>
> **Therefore, the variance is well-managed and does not compromise the core complexity claim; the quadratic scaling advantage over classical RRL.**

---

> > ### Comment · Reviewer_29uc · 2025-11-26
> >
> > Thank you for answering all my questions. I will maintain my positive rating of the submission.

---

> > > ### Author Response · Authors · 2025-11-26
> > >
> > > Thank you for your time and positive feedback. We sincerely appreciate your constructive review, which helped us improve the paper.

---

### Official Review · Reviewer_zi4K · 2025-11-01

**Soundness:** 2
**Presentation:** 2
**Contribution:** 2
**Rating:** 2
**Confidence:** 3

**Summary:**

This paper explores the advantages of modeling environment interactions in reinforcement learning using quantum computing concepts. The authors formalize uncertainty in robust RL objectives through a quantum-inspired representation based on superposition, which is used to accelerate the inner minimization problem and potentially improve sample efficiency in quantum-compatible environments.

The proposed approach aims to extend quantum reinforcement learning (QRL) beyond the design of quantum-enhanced policies, suggesting that environmental uncertainty and stochastic dynamics can also be modeled using quantum phenomena. This is an intriguing idea that could bridge the conceptual gap between quantum mechanics and environment modeling in RL.

**Strengths:**

- The paper engages with a timely and conceptually rich topic, namely the use of quantum principles for modeling environmental uncertainty in RL.
- The proposed formulation could help frame future discussions on quantum-compatible environment representations and robustness in hybrid quantum-classical learning setups.
- The theoretical framing is reasonably well-motivated, even if the presentation is difficult to follow.

**Weaknesses:**

- Incremental novelty: The core idea builds upon earlier proposals to apply quantum concepts to environment modeling. The contribution here is a modest extension, not a genuinely new paradigm.
- Limited empirical support: Experimental results are largely comparable between classical RL and classical robust RL. The claimed benefits (improved robustness, faster convergence) are not demonstrated convincingly.
- Poor organization: The structure obscures the motivation and significance of the method. The overall narrative could benefit from clearer framing and reordering.
- Lack of illustrative examples: A small, well-chosen toy example could clarify the proposed mechanism and highlight the degradation the authors aim to mitigate.
- Missing discussion of limitations: The paper does not adequately address the scalability challenges or hardware constraints of the proposed approach.

**Questions:**

1. Hardware applicability: Is the approach intended for near-term NISQ devices or future fault-tolerant architectures?
2. Noise modeling: Were the reported experiments noise-free simulations or did they incorporate realistic hardware noise models?
3. Demonstrated advantage: Can the authors show a specific setting where the proposed uncertainty representation yields a measurable policy improvement?

---

> ### Author Response · Authors · 2025-11-20
> **Response to Reviewer zi4K (Part 1/4)**
>
> ### **Dear Reviewer zi4K**,
> We sincerely appreciate your thoughtful and constructive comments. We thank you for the opportunity to respond to your review and hope this relieves your concern.
>
> &nbsp;
>
> >### **Weakness 1. Incremental novelty**
>
> We agree that our work has been built on the existing concepts of Quantum-Accessible Environments, Quantum Amplitude Estimation (QAE), and Quantum Minimum Finding (QMF).
>
> We hope to emphasize that **our core novelty lies in formulating and addressing a key bottleneck in Robust Reinforcement Learning (RRL) through a quantum perspective.**
>
> Classical RRL methods suffer from the extensive inner-loop minimization, whose query complexity scales up with $\mathcal{O}(|\mathcal{U}|)$ for finding the worst-case scenario, *a critical bottleneck from the deep-learning (DL) field.*
>
> Our work is the first to frame this specific RRL bottleneck as a quantum search problem. By using QAE to estimate the 1-step returns and QMF to find the minimum, our QRIM algorithm achieves a provable quadratic speed-up with $\mathcal{O}(\sqrt{|\mathcal{U}|})$ in this inner loop.
>
> Notably, for Cartpole cases, this speed-up leads to **-79.5% reduction in queries** compared with classical RRL, also yielding **`significant' acceleration of training** (performance vs. query counts curve). We added details in the response to Weakness 2.
>
> Along with the considerable gains in empirical testing, we believe applying these quantum concepts to address the sample-scalability bottleneck in the practical domain of DL is NOT straightforward and **offers an interdisciplinary perspective that combines a DL perspective with the quantum world.** We hope to raise the possibility of DL+Quantum breaking the huddle over the *scalability-generalization issue in modern AI*.

---

> ### Author Response · Authors · 2025-11-20
> **Response to Reviewer zi4K (Part 2/4)**
>
> >### **Weakness 2. Limited empirical support**
>
> Let us first clarify two distinct performance claims of our algorithm QRIM (Quantum RRL): (1) Robustness gain over the classic counterparts, and (2) Quadratic reduction in query complexity over Classic RRL.
>
> ### **Robustness**
>
> Regarding robustness, we hope to clarify that our primary goal is to reduce the number of queries while achieving robust policies. Thus, QRIM aims to achieve robustness comparable to that of classical RRL, beyond that of non-robust classical RL.
>
> **(1) Classical RRL trivially outperforms classical RL (no doubts on the efficacy of RRL):**
> - Let us focus on the part of the test distribution with a significant deviation from the train (dubbed ‘extrapolation’ in our paper).
> - In FrozenLake extrapolation cases (Slip prob. $\rho\geq 0.10$, hole index $\geq 5$; in Fig. 3a, 3b), classical RRL ($\textcolor{orange}{orange}$ curves) definitely outperform classical RL ($\textcolor{blue}{blue}$ curves).
> - In CartPole extrapolation cases (Pole mass $\geq 0.22$ OR $\leq 0.08$, Joint Damping $\geq 0.12$ OR $\leq 0.03$; in Fig. 5a, 5b), classical RRL is generally more robust than classical RL, and, in some cases, it is at least comparable.
>
> **(2) QRIM (quantum RRL) outperforms its classical counterparts:**
> - (In Fig 3, 5) $\textcolor{green}{Green}$ curves of QRIM positions on top of $\textcolor{orange}{orange}$ (classical RRL) and blue (classical RL)) curves almost everywhere (QRIM achieves at least better robustness than classical RRL).
> - Beyond the plot-based visualizations, we hope to add numerical results on the performance of QRIM, classical RRL, and classical RL (see the following Table).
> - QRIM apparently outperforms its classical counterparts, while showing less performance reduction percentiles from the shifting of train $\rightarrow$ test (extrapolation). We kindly inform that the numbers in the following table are already included in the robustness Table of our paper (Table 3).
> - We hope these numerical results relieve your concern.
>
> |  | **Classical RL** |  | **Classical RRL** |  | **Quantum RRL (QRIM)** |  |
> | --- | --- | --- | --- | --- | --- | --- |
> |  | CartPole | FrozenLake | CartPole | FrozenLake | CartPole  | FrozenLake |
> | train  (mean) | 61.95 | 0.76 | 61.03 | 0.84 | **76.41** | **0.98** |
> | test  (mean) | 26.65 | -0.79 | 39.86 | -0.45 | **58.06** | **-0.37** |
> | degradation(-(train-test)/train * 100) | -56.98% | -203.94% | -34.68% | -153.57% | **-24.02%** | **-137.75%** |
>
> ### **Faster Convergence**
>
> QRIM’s speed-up benefits indeed lead to faster training; less querying trivially yields faster convergence. In the submitted paper, we showed the scale-up behavior for uncertainty size to precisely validate our claim. We hope to present the actual benefits of speed-up.
>
> **We additionally measured and compared the total cumulative number of oracle (query) calls throughout the entire training process.** For fairness, classical RRL and QRIM execute the same number of training iterations (i.e., FrozenLake: 1,000 and CartPole: 200 iterations).
>
> As shown in the following table, QRIM exhibits a substantial reduction in total query counts. In FrozenLake and CartPole, **QRIM only needs 35.9% and 20.5% of the query counts of classical RRL**, respectively. We have added these results to the revised paper (Section 5.4).
>
> |  | **Uncertainty grid ($N$)** | **Classical RRL** | **Quantum RRL (QRIM)** |
> | --- | --- | --- | --- |
> | FrozenLake | 4 | 11.45M | **4.11M** |
> | CartPole | 36 | 27.51M | **5.65M** |
>  \*  'M' indicates $10^6$
>
> **We also provide the learning curves of performance vs. number of queries for both benchmarks (added to Section 5.4, Fig. 7).** In the plots, we can clearly observe the speed-up of training for QRIM beyond classical RRL, empirically validating QRIM’s speed-up benefits. For FrozenLake, QRIM reaches almost-saturated performance at 25K queries, but classical RRL needs almost 125K queries to show a sufficient performance. For CartPole, QRIM shows a rapid performance increase even with 250K queries, but classical RRL still sticks to the almost-random performance (just beginning to learn a meaningful policy), even after pouring more than 200K queries ('K' indicates $10^3$).

---

> ### Author Response · Authors · 2025-11-20
> **Response to Reviewer zi4K (Part 3/4)**
>
> >### **Weakness 3,4. Poor organization, Lack of illustrative examples**
>
> As the reviewer pointed out, we aim to highlight the key parts of our paper, with a particular emphasis on robustness and speed-up benefits of QRIM. We added the following revisions:
>
> **A case illustration of speed-up gains:** First, we have added an illustrative example to the Introduction (referring to Fig. 1). We hope our readers catch the key claim at the beginning of this article. Specifically, we use a simple "toy" MDP from FrozenLake (e.g., $|\mathcal{U}|=4$) to intuitively walk the reader through $\mathcal{O}(4)$ classical scan vs. $\mathcal{O}(\sqrt{4})$ quantum search. We believe this will help readers catch the key idea and benefit, along with the following verbal descriptions in the initial submission:
> - L58~61: “our proposed quantum robust inner minimization (QRIM) ~ whose query complexity is  $\mathcal{O}(\sqrt{\mathcal{U}})$ which is significantly lower than the conventional worst-case exhaustive search (i.e., $\mathcal{O}(\mathcal{U})$ of RRL).”
>
> **A special emphasis on speed-up results (robustness and query counts):** In the Introduction, we have added a brief comment on robustness and speed-ups with gain numbers. The revision of the corresponding part is as follows:
> - L84-87: “Lastly, based on a classical emulator for quantum-access oracle, we confirm that QRIM learns robustness policies with significantly reduced total query counts, e.g., it achieves ×2.17 out-of-distribution test reward over classical RL, and −79.5% query reduction compared to classical RRL in the CartPole benchmark.”
>
> **Adding more empirical results (corresponding to the response to Weakness 2):** We have added query counts results, validating the speed-up gains of QRIM (Table 4). Also, we newly added the full learning curves (reward vs. query counts) of classical RRL and QRIM (Fig. 7).
>
> We hope this revision addresses your concern about the presentation.

---

> ### Author Response · Authors · 2025-11-20
> **Response to Reviewer zi4K (Part 4/4)**
>
> >### **Weakness 5 & Question 1,2. Missing discussion of limitations, Hardware applicability, Noise modeling Limited empirical support**
>
> **Quantum Hardware applicability:** In our paper, the reported experiments are based on noise-free simulations. Also, our components, QAE and QMF, generally assume *noise-free* fault-tolerant (FT) architectures.
>
> Quantum hardware validation is a direct way to demonstrate the scalability and practicality of the idea. We acknowledge that this is particularly important in the quantum world.
>
> However, **our initial goal was to theoretically figure out how a quantum nature brings quadratic query speed-up to robust deep learning. We have tried our best to extend our theory to quantum hardware in the rebuttal.**
>
> *We are indeed conducting a small-scale demonstration on ‘real’ quantum hardware for the FrozenLake environment, thanks to the unexpected quantum resources we can now temporarily use.* It is not guaranteed, but we sincerely hope to report the results within the rebuttal period. Also, we ask for your kind understanding that the quantum hardware is still NOT prevalent, and the current commercialized cloud services are extremely costly (hard to support with our research funding).
>
> **Noise modelling:** Thanks to your suggestion, **we additionally tested QRIM with noise modelling, i.e., Qiskit Aer noise model** (with depolarizing error rates of 0.05% and 0.6% for 1-Qubit and 2-Qubit error, respectively, and a 2% readout error).
>
> We found that QRIM consistently shows its superiority even in the noise model with **minimal changes in performance (reward)**, with only **4% increase in total query counts** (still much fewer queries and better robustness than classical RRL).
>
> We have added this result to the revised manuscript (Appendix G).
>
> |  | **QRIM noise-free** | **QRIM w/ noise** |
> | --- | --- | --- |
> |  | FrozenLake | FrozenLake |
> | train | 0.98 | 0.91 |
> | test | -0.37 | -0.44 |
>
>
> **Additional Discussions on Noisy/Real Quantum Environments:** In addition, our theory provides an intuition about its practicality in noisy quantum hardware.
>
> In the scale behavior of QRIM, i.e., $\tilde{\mathcal{O}}(\sqrt{N} \log(1/\delta))$ (Prop. 3), $\delta$ means the failure probability for minimum search. In our noise-free simulations, we already set an imperfect search with a success probability of $(1-\delta)=0.90<1$. It reflects the stochastic behavior of quantum environments.
>
> In the literature [4], hardware noise is shown to degrade the success probability, which is another factor that might affect the success of minimum search.
> - [4] Hein et al. “From Classical Data to Quantum Advantage – Quantum Policy Evaluation on Quantum Hardware” (2025)
>
> As we already assume imperfect minimum search, i.e., ($1-\delta)=0.90$, in our experiments, QRIM’s successful training even with such imperfect search makes us conjecture QRIM’s possible robustness against hardware noise, which also disturbs perfect search.
>
> This leads us to conjecture that QRIM shows sufficient robustness in noise-model-based experiments, coinciding with the previous noise-model results. Also, if we set a severe failure probability (a high $\delta$) for QRIM, it may lead to a slight increase in the required query counts to keep the robustness performance.
>
> We have added this discussion to the limitations section in the Appendix G, clarifying QRIM’s practicality in a real quantum environment.
>
> &nbsp;
>
> >### **Question 3. Demonstrated advantage**
>
> As detailed in our response to Weakness 2, we hope to kindly emphasize that QRIM’s measurable policy improvement has been observed in FrozenLake and CartPole benchmarks.

---

> ### Author Response · Authors · 2025-11-26
>
> ### **Dear Reviewer zi4K**,
>
> Following your feedback, we have successfully conducted experiments on a real quantum hardware of 127-qubit IBM Eagle processor.
> Please refer to our general response(Additional Results: Real Quantum Hardware Verification) and the newly added Appendix G for the comprehensive results.

---

> > ### Comment · Reviewer_zi4K · 2025-11-27
> >
> > Thank you for the clear and thorough rebuttal. Most of my earlier concerns have been successfully addressed:
> > - The added robustness metrics, query-efficiency results, and learning-curve plots now convincingly demonstrate both the empirical benefits and the practical impact of the quadratic speed-up.
> > - The introduction and new illustrative example greatly improve clarity.
> > - The added discussion of hardware assumptions, noise-model and real-hardware experiments provides additional context on applicability.
> >
> > A few minor points remain:
> > - The contribution is still largely an application of known quantum primitives, though you now clearly acknowledge this.
> > - The weaker worst-case results in CartPole and the general interpretation of Figs. 2 and 4 would benefit from a brief explanation in the main text.
> > - A short limitations paragraph (hardware assumptions, overhead and implications of the hybrid setup, ...) in the main body would improve transparency.
> >
> > Overall, the revisions significantly strengthen the paper. I will update my score accordingly.

---

> ### Author Response · Authors · 2025-11-28
>
> We sincerely appreciate your detailed and thorough feedback to improve our work further. We are glad to hear that our additional experiments and revisions have successfully addressed your major concerns.
>
> Regarding the remaining minor points, we have carefully reflected them in the manuscript to further strengthen clarity and transparency:
>
>
> >### **Minor point 1: The contribution is still largely an application of known quantum primitives**
>
> As you pointed out, we acknowledge that our work builds upon known quantum primitives.
>
> Our work first examines the possibility of quantum principles for guiding robust RL, which offers significant potential for query scalability in DL.
>
> As suggested, we hope to leave the novel design of quantum primitives tailored for robust DL dynamics as our near-future work, and we are confident that this direction will have a significant impact on both DL and quantum communities.
>
>
> >### **Minor point 2: The weaker worst-case results in CartPole**
>
> We have added our brief explanations in **Section 5.3** to clarify the results of FrozenLake and Cartpole. Also, regarding the weaker worst-case results in CartPole, we attribute this to the probabilistic nature of the quantum algorithm in solving the Max-Min problem. Given that QRIM currently utilizes significantly fewer queries than classical RRL, we can allow more queries to achieve higher confidence in the quantum-based Max-Min solver, which is a possible remedy.
>
> >### **Minor point 3: Limitations paragraph**
>
> We reformed the concluding part as **"Conclusion and Limitations"** to explicitly include the limitations in the main text. We have added a dedicated discussion summarizing the constraints of the hybrid setup and the requirements for full quantum oracles, as follows.
>
> Again, we truly appreciate your constructive comments, letting us significantly improve our work. Also, we kindly request your final update based on your assessment of the revised paper. Thank you.

---

### Author Response · Authors · 2025-11-20
**Overview of Manuscript Revisions**

Dear reviewers,

We sincerely appreciate your thorough, constructive, and insightful feedback. We have carefully revised the manuscript to improve clarity, strengthen technical explanations, and address all weaknesses and questions based on your comments. All revised text in the manuscript has been marked in $\textcolor{blue}{blue}$ for clarity.

### **For Reviewer zi4K**

- We have added $\textcolor{blue}{Table 4}$ to show that QRIM exhibits a substantial reduction in total query counts than classic RRL **(Weakness 2,3)**.
- We have added $\textcolor{blue}{Figure 7}$ to provide the learning curves of performance vs. number of queries of classic RRL and QRIM (**Weakness 2,3**).
- We have added $\textcolor{blue}{Figure 1}$ to present an illustrative example of speed-up gains of QRIM over classic RRL (**Weakness 3,4**).
- We have revised $\textcolor{blue}{L58-61}$, &  $\textcolor{blue}{L84-87}$ to help readers catch the key idea and benefit of QRIM(**Weakness 3**).
- We have added $\textcolor{blue}{Appendix G}$ discuss about the limitations of quantum hadware verification, and address the concerns of QRIM regarding hardware applicability with the test of noise modeling (**Weakness 5 & Question 1,2**)

### **For Reviewer 29uc**

- We have corrected typos in  $\textcolor{blue}{L164, L166, L537}$ (**Weakness 3**)
- We have revised unnatural expressions in $\textcolor{blue}{L192, L357}$ (**Weakness 3**)
- We have added the definition of the symbol used in  $\textcolor{blue}{Definition 1}$ (**Question 1**)
- We have added  $\textcolor{blue}{Appendix D}$ to give detailed explanations of qubits used for amplitude encoding (**Question 2**)

### **For Reviewer YhQS**

- We have added  $\textcolor{blue}{Appendix D}$ to give detailed construction of quantum environment oracles and circuit descriptions for the benchmark environment (**Weakness 1 & Question 1**).
- We have rephrased $\textcolor{blue}{L341-343}$ for better clarity (**Weakness 2**)
- We have corrected typos in $\textcolor{blue}{L508}$ (**Weakness 2**)

### **For Reviewer YZjp**

- We have added $\textcolor{blue}{Appendix G}$ discuss about the limitations of quantum hadware verification, and address the concerns of QRIM regarding hardware applicability with the test of noise modeling (**Weakness 1**)
- We have corrected typos in $\textcolor{blue}{L164, 166}$ (**Weakness 2**)
- We have corrected typos in $\textcolor{blue}{References}$ (**Weakness 2**)
- We have added a reference in $\textcolor{blue}{L113}$ (**Question 2**)

Once again, we sincerely thank all reviewers for your valuable comments and constructive discussions. Your feedback has significantly strengthened the clarity, rigor, and overall quality of our work. We truly appreciate your time and effort in reviewing our manuscript.

---

### Author Response · Authors · 2025-11-26
**Additional Results: Real Quantum Hardware Verification**

Dear reviewers,

To address the concerns regarding the feasibility of our approach on physical quantum devices (particularly inquired by **Reviewers zi4k**, **YZjp**), we have conducted additional experiments using real quantum hardware (IBM Quantum backend ‘ibm_*anonymous institute*’ with 127-qubit Eagle processor).

We have conducted real quantum hardware experiments on the FrozenLake 8x8 environment using the same quantum RRL setup described in the Experiment section (we used 2 qubits and 20 depth circuits to implement QRIM inner-minimization).

As shown in the table below, QRIM executed on real quantum hardware achieved an extrapolation return of -0.52, which is slightly lower than the Classic RRL (-0.45) and the noise-free QRIM (-0.37) due to hardware noise, **but significantly outperforms Classic RL(-0.79).**

The results confirm that QRIM maintains the performance trend observed in simulations, demonstrating practical robustness even on noisy intermediate-scale quantum devices (NISQ).

|  | QRIM(noise-free) | QRIM(noise modeling) | **QRIM(quantum hardware)** | Classic RL | Classic RRL |
| --- | --- | --- | --- | --- | --- |
| train | 0.98 | 0.91 | **0.94** | 0.76 | 0.84 |
| test(extrapolation) | -0.37 | -0.44 | **-0.52** | -0.79 | -0.45 |

Also, the quadratic query speed-up ($\mathcal{O}(\sqrt{\mathcal{U}})$) was preserved on real quantum hardware. **The total number of queries had been increased by only 5% compared to the noise-free QRIM, which is still much fewer queries than classical RRL.**

These new results demonstrate that QRIM outperforms classical RL and maintains robustness trends even under real hardware noise, validating the practical potential of our method beyond theoretical advantages.

We have updated the manuscript to include the results in Appendix G.

---

### Author Response · Authors · 2025-12-02
**Summary for the New Area Chair: Review Status & Key Revisions**

Dear Area Chair,

Given the discussion reset following the recent incident, and since our rebuttal and manuscript revisions became substantial, we summarize the discussions of the reviews and how the main concerns have been resolved in the revised manuscript to help your decision.

>### **Summary of the discussion review status**


- **Reviewer 29uc (Rating 8):** **Explicitly confirmed maintaining a positive assessment** after our response.
- **Reviewer YZjp (Rating 8):** No further comments after our response
- **Reviewer YhQS (Rating 4):** No further comments after our response.
- **Reviewer zi4K (Rating 2):** Later stated in the final official comment that “**most concerns were resolved and intend to update the score accordingly**.”

>### **Key revisions addressing the major concerns**


- **Hardware applicability (zi4k, YZip):** We addressed the real quantum hardware feasibility concern by executing QRIM on NISQ device (IBM Eagle 127-qubit processor). Results in **Appendix G** confirm robustness under real hardware noise. *Reviewer zi4K acknowledged that this addition significantly strengthened the paper.*
- **Empirical support of robustness and query efficiency (zi4K, 29uc):** We added **Table 4** and **Figure 7**, demonstrating QRIM’s robust performance while substantially reducing query complexity compared to classical RRL. *This directly resolved the concern on empirical benefits, which were positively acknowledged by reviewers.*
- **Quantum oracle construction (YhQS):** We added **Appendix D** with detailed circuit constructions (transition/reward encodings), *resolving the reviewer’s main weakness regarding detailed explanations of the oracles.*
- **Novelty and relation to prior work (zi4K):** We clarified our contribution in the **Introduction** and the **Conclusion with limitations** section that *our core novelty lies in formulating and addressing a key bottleneck in robust reinforcement learning as a quantum minimum-search problem,* rather than proposing new quantum primitives.
- **Organization and illustrative example (zi4K):** We addressed the concern about organization and lack of examples, adding **Figure 1** and a toy example in the **Introduction** to intuitively illustrate the query complexity advantage. *Reviewer zi4K acknowledged that this  significantly improved the presentation.*

We respectfully hope that the above summary is helpful for your evaluation, and we sincerely appreciate your time and consideration.

---

### Note · Program_Chairs · 2026-01-17
**Submission Desk Rejected by Program Chairs**

The following references in this submission do not refer to real documents and/or have major errors in bibliographic information:

 Yudi Chen, Rui Gao, and Yao Xie. Robust Markov decision processes: Beyond rectangularity. In Proceedings of the 36th International Conference on Machine Learning (ICML), volume 97 of Proceedings of Machine Learning Research, pp. 1142-1151. PMLR, 2019.